# Gut Microbiome and Immune System Crosstalk in Chronic Inflammatory Diseases: A Narrative Review of Mechanisms and Therapeutic Opportunities

**DOI:** 10.3390/microorganisms13112516

**Published:** 2025-10-31

**Authors:** Jefferson J. Feng, Nikhil R. Maddirala, Ashley Saint Fleur, Fenfen Zhou, Di Yu, Feng Wei, Yongrong Zhang

**Affiliations:** 1Department of Microbial Pathogenesis, School of Dentistry, University of Maryland, Baltimore, MD 21201, USA; jfeng3855@gmail.com (J.J.F.); maddiralanikhilr@gmail.com (N.R.M.); asaintfleur@umaryland.edu (A.S.F.); fzhou@umaryland.edu (F.Z.); dyu1@umaryland.edu (D.Y.); 2Department of Neural and Pain Sciences, School of Dentistry, University of Maryland, Baltimore, MD 21201, USA; fwei@umaryland.edu

**Keywords:** gut microbiome, dysbiosis, chronic inflammation, pathogenesis, immune response, rheumatoid arthritis, inflammatory bowel disease, psoriasis, systemic lupus erythematosus, asthma, vasculitis

## Abstract

The gut microbiota, a complex community of trillions of microorganisms residing in the gastrointestinal tract, plays a vital role in maintaining host health and regulating a wide range of physiological functions. Advances in molecular biology have greatly expanded our understanding of the dynamic interactions between the gut microbiome and the immune system. Disruption of this microbial community, known as dysbiosis, can compromise epithelial barrier integrity, trigger aberrant immune activation, and lead to the production of proinflammatory metabolites. These changes are increasingly recognized as contributing factors in the pathogenesis of chronic inflammatory diseases. Emerging research highlights the gut microbiota as a key modulator of immune homeostasis, influencing both local and systemic inflammatory processes during the initiation and progression of these diseases. Understanding the mechanisms underlying gut microbiota-immune interactions will offer new avenues for therapeutic interventions. This review focuses on six representative chronic inflammatory diseases, including rheumatoid arthritis, inflammatory bowel disease, psoriasis, systemic lupus erythematosus, asthma, and vasculitis, all of which are characterized by dysregulated immune responses and persistent inflammation. Our goal is to synthesize the recent research on the role of gut microbiome in the pathogenesis of the diseases listed above and provide insights into the development of microbiota-based therapies, particularly fecal microbiota transplant, dietary modifications, prebiotic and probiotic interventions, for their treatment.

## 1. Introduction

Chronic inflammatory diseases, encompassing a wide range of conditions, are characterized by persistent inflammation in various tissues and organs, resulting in tissue damage, dysfunction, and severe clinical symptoms. Despite notable advances in molecular biology and a deeper understanding of pathophysiology, these diseases remain a significant and unresolved problem. The gut microbiota has emerged as a central figure in the pathogenesis and progression of chronic inflammatory diseases. Research over recent decades has uncovered the complex relationships between the alteration in the gut microbiota and both the development and perpetuation of chronic inflammation. Dysbiosis, characterized by alterations in the composition and function of the gut microbiota, has been implicated in chronic inflammatory diseases [1] (Table 1).

One of the primary mechanisms through which the gut microbiota contributes to chronic inflammation is modulating mucosal immune responses. Immune cells and intestinal epithelial cells interact with gut microbes and their products, shaping immune development, responsiveness, and tolerance [2]. Gut microbiome is also essential for maintaining the intestinal barrier integrity thereby preventing the translocation of pathogens and proinflammatory stimuli into systemic circulation. When dysbiosis occurs, this delicate balance between host and gut microbe is disrupted, triggering aberrant immune activation, inflammation and tissue damage [3]. Emerging evidence suggests that the gut microbiota communicates bidirectionally with distant organs and tissues via systemic immune responses and neuroendocrine signaling pathways. This crosstalk influences the pathogenesis of extraintestinal chronic inflammatory diseases, such as rheumatoid arthritis (RA), psoriasis, and asthma, highlighting the intimate link between the gut microbiota, systemic inflammation, and immune-mediated disorders [4]. Additionally, the gut microbiota produces a myriad of metabolites, including short-chain fatty acids (SCFAs), bile acids, lipopolysaccharides (LPS), and microbial antigens, which are capable of modulating immune cell function and inflammatory signaling pathways [5].

In light of these findings, there is growing interest in targeting the gut microbiota and its metabolites as a therapeutic strategy for chronic inflammatory diseases. This article aims to explore the intricate functions of the gut microbiome within the human body, with a particular emphasis on its pivotal role in the onset and progression of several major chronic inflammatory conditions. Specifically, we discuss RA, inflammatory bowel disease (IBD), psoriasis, systemic lupus erythematosus (SLE), asthma, and vasculitis, diseases that collectively affect millions of individuals worldwide and pose a substantial global health burden. We also discuss the therapeutic potential of microbiota-targeted intervention, including dietary modifications, probiotics, prebiotics, and fecal microbiota transplantation (FMT) which were among the most frequently and prominently retrieved interventions when searching with the keywords listed below. To ensure relevance and currency, we reviewed articles written in English from the past 15 years, identified through the PubMed database using the keywords correlated to the topics of “microbiome”, “chronic inflammatory diseases”, “host immune response and gut microbes”, “host immune response and microbial metabolites” and the specific disease names listed above. Our goal is to synthesize current knowledge of gut microbiome-related pathogenesis, highlight mechanistic insights, and identify therapeutic opportunities across those major disease areas.

## 2. Overview of Gut Microbiome-Driven Immune and Inflammatory Modulation

About 100 trillion microorganisms consisting of bacteria, fungi, archaea, protozoa, and viruses, reside in the gastrointestinal (GI) tract, forming the complex community known as the gut microbiome [6,7]. This ecosystem maintains a mutually beneficial relationship with its host. The host supplies those microorganisms with nutrients and a protected habitat, while the microbiota aids in digesting the dietary components, communicates with the immune system, and shapes overall host immunity. When this symbiotic balance is disturbed, often referred to as dysbiosis, various diseases may develop, including infections, neurological disorders, chronic inflammation, and metabolic syndromes. For example, following antibiotic treatment that depletes many commensals, the opportunistic pathogen *Clostridioides difficile* can thrive to cause life-threatening colitis. We have defined terms including “dysbiosis”, “microbiota”, “microbiome”, “disease driver”, “biomarker” and “pathway” in Table 1 [8].

Dysbiosis alters the profile of gut-derived metabolites, which constitute another crucial component of the microbiome. These small molecules interact with the intestinal mucosal immune system, influencing antigen recognition, immune cell recruitment and proliferation. Gut microbiota-derived metabolites are generally categorized into three main groups: (1) direct microbial products of dietary compounds (e.g., SCFAs, tryptophan and its indole derivatives); (2) host-derived molecules chemically modified by the microbiota (e.g., secondary bile acids); and (3) metabolites synthesized de novo by gut microbiota (e.g., LPS and peptidoglycan) [9]. By modulation of immune-cell metabolism and signaling pathways, these metabolites profoundly impact on inflammation and overall host health [10,11].

In this section, we briefly discuss the mechanisms by which the gut microbiota and its metabolites modulate host inflammatory response (Figure 1).

### 2.1. Gut Epithelial Barrier

Gut epithelium is the very first-line defense against invasive pathogens and their virulent factors in the GI tract. A healthy gut barrier prevents the translocation of various microbial antigens and inflammatory substances into the systemic circulation, thereby reducing the risk of systemic inflammation and immune activation [12,13,14].

Normally, beneficial microbes and their metabolites play essential roles in protecting and maintaining gut epithelial barrier functions. Reduced levels of key beneficial gut bacteria, such as *Bifidobacterium* and *Lactobacillus*, which help strengthen the gut barrier, are often linked to increased gut permeability [15,16]. Urolithin A (UroA), a direct microbial product of dietary compounds, and its synthetic analogs enhance gut barrier function through the nuclear factor erythroid 2-related factor 2 (Nrf2) pathway, leading to the inhibition of unwarranted inflammation [17]. SCFAs contribute to maintenance of the integrity of the intestinal epithelial barrier by upregulating the expression of tight junction proteins [18]. SCFAs also play a key role in modulating antitumor immune responses and reducing inflammation-related side effects by supporting epithelial barrier function and preserving intestinal homeostasis [19]. Specific secondary bile acids, such as ursodeoxycholic acid (UDCA) and tauroursodeoxycholic acid (TUDCA), have shown promise in reducing colitogenic dysbiosis and suppressing experimental colitis [20]. However, dysbiosis may lead to the suppression of the beneficial microbes and the outgrowth of pathogens. In such scenarios, pathogens and their virulence factors, such as *C. difficile* toxins, shiga toxins, and enterotoxins, disrupt the barrier function and induce strong local inflammatory responses. With increased epithelial permeability, the de novo microbial metabolites further contribute to systemic inflammation. One of the classic systemic inflammation inducers is LPS. Gut barrier dysfunction leads to increased permeability to bacterial LPS, which triggers immune responses and subsequently contributes to chronic inflammation [21]. Emerging research also reveals the role of microbiota-derived extracellular vesicles (BEVs) in modulating immune and defense responses and potentially contributing to the development of diseases through affecting the intestinal epithelial barrier [22,23].

### 2.2. Interactions with Immune Cells

The gut microbiota plays a pivotal role in modulating the host immune system by directly interacting with various immune cells, such as dendritic cells (DCs), macrophages, T cells and B cells, thereby influencing their development, differentiation, and function. As primary antigen-presenting cells, DCs are essential for initiating and regulating immune responses. Microbial metabolites, such as SCFAs and secondary bile acids, modulate DC maturation and cytokine production, which in turn affect T cell differentiation and promote immune tolerance. The gut microbiota also influences macrophages polarization, directing them towards either the proinflammatory M1 phenotype or the anti-inflammatory M2 phenotype [24,25]. SCFAs, particularly butyrate, promote the M2 phenotype, which is associated with tissue repair and anti-inflammatory functions [26]. Additionally, microbial metabolites like butyrate have a significant impact on T cell differentiation and function [27]. Butyrate promotes the differentiation and expansion of regulatory T cells (Tregs) by inhibiting histone deacetylases (HDACs) [28,29], leading to increased acetylation of histone proteins and the expression of genes involved in Treg differentiation. Tregs are essential for maintaining immune tolerance and preventing excessive inflammation. Furthermore, the gut microbiota modulates B cell function and antibody production [30]. Microbial antigens can promote the production of immunoglobin A (IgA), which plays a crucial role in mucosal immunity by neutralizing pathogens and preventing their adherence to the intestinal epithelium [31]. However, deleterious interactions also occur. As we will discuss in relation to specific diseases later, molecular mimicry of bacterial antigens may trigger the production of autoantibodies that mistakenly target host antigens, contributing to autoimmune pathology.

#### 2.2.1. Toll-Like Receptors

Pattern recognition receptors (PRRs) on immune cells, such as Toll-like receptors (TLRs), play critical roles in recognizing microbial-associated molecular patterns (MAMPs) and initiating immune responses [32]. The interaction between the gut microbiota and TLRs serves as a key mechanism of immune modulation.

TLR4 recognizes LPS from Gram-negative bacteria [33]. LPS is a potent immune modulator that can activate immune cells via TLR4 and induce massive inflammatory responses or regulatory responses, depending on the context and balance of other signaling molecules [34]. Chronic activation of TLR4 by the dysbiotic gut microbiome contributes to systemic inflammation and autoimmune diseases [35]. TLR5 recognizes flagellin, a component of bacterial flagella [36]. The interaction between TLR5 and flagellin can promote the production of anti-inflammatory cytokines, such as IL-10, and support the integrity of the gut barrier. TLR5 signaling is essential for maintaining a balanced immune response and preventing excessive inflammation [37]. TLR2 recognizes peptidoglycan and other components of the cell wall of Gram-positive bacteria [38]. TLR2 activation can increase the production of antimicrobial peptides and promote mucosal immunity [39]. Additionally, TLR2 signaling can modulate the balance between T helper cell subsets, influencing the overall immune response [40]. Hug et al. [41], have reviewed the expression and localization of TLRs in the GI tract and highlights their role in maintaining gut barrier function and the implications of disrupted signaling in inflammatory conditions such as IBD.

#### 2.2.2. Proinflammatory and Anti-Inflammatory Cytokines

The gut microbiome can not only modulate immune cell maturation but also induces the production of cytokines, thereby directing immune responses [8]. For example, certain beneficial bacteria such as *Lactobacillus* can promote the production of IL-10 by DCs, leading to the induction of Tregs and the suppression of inflammatory responses [42]. SCFAs can also influence cytokine production. Propionate and butyrate can suppress the production of proinflammatory cytokines such as tumor necrosis factor-alpha (TNF-α), interleukin (IL)-6, and IL-12 by inhibiting the activation of the nuclear factor kappa B (NF-κB) and mitogen-activated protein kinase (MAPK) signaling pathways [43]. This modulation contributes to reducing systemic inflammation and promotes immune homeostasis.

Overall, the gut microbiota and their metabolites exert a profound influence on the host immune system, by affecting various immune cells and their functions. Understanding these interactions provides valuable insights into maintaining immune balance and developing novel therapeutic strategies for chronic inflammatory diseases.

## 3. Gut Microbiota and Chronic Inflammatory Diseases

### 3.1. Rheumatoid Arthritis (RA)

#### 3.1.1. General Understanding of RA Pathogenesis

RA is a chronic autoimmune disorder characterized by inflammation of the synovial joints, resulting in pain, swelling, stiffness, and progressive joint damage. RA impacts approximately 0.5–1% of the adult population. While the exact cause of RA remains elusive, it is widely believed to arise from a complex interplay of genetic, immunological, and environmental factors. Genetic susceptibility plays a significant role, with certain alleles within the human leukocyte antigen (HLA) region, particularly the HLA-DRB1 gene, strongly associated with increased risk (Table 2) [44]. The dysregulated immune system of RA patients leads to the extensive activation of immune cells, including T cells, B cells, and macrophages. Meanwhile, autoantibodies, such as anti-rheumatoid factor and anti-citrullinated protein antibodies (ACPAs) are significantly produced [45]. This immune activation mistakenly attacks the patient’s own tissues and elicits chronic inflammation and synovitis, consequently causing cartilage and bone destruction, joint deformity, and disability if left untreated. Moreover, RA is not limited to joint involvement; it is frequently associated with systemic manifestations such as fatigue, fever, weight loss, and an increased risk of cardiovascular disease and osteoporosis [46].

#### 3.1.2. Microbiome and RA Pathogenesis

Recent studies have revealed that microbial composition changes as RA progresses in patients, which suggests that certain microbes may either protect against or contribute to the pathogenesis of the disease [69]. The role of the gut microbiota in RA is a rapidly growing area of research, reflecting increased recognition of the gut-joint axis in autoimmune diseases. The interplay between the gut microbiota and host genetics, environmental factors, and dietary habits further complicates the relationship between gut dysbiosis and RA susceptibility. Environmental factors, smoking and dietary choices, have been shown to influence the composition of the gut microbiota and its metabolites, further contributing to the complexity of RA pathogenesis [70].

Although arguments persist about whether the observed microbial changes in RA patients are a consequence of disease or a causative factor, substantial evidence has demonstrated that microbiota actively participates in RA pathogenesis. Several studies have highlighted specific bacterial species, such as *Prevotella copri* and *Porphyromonas gingivalis*, in the onset and progression of RA [71]. *P. copri*, for example, has been linked to early synovitis and increased disease severity [72], whereas *P. gingivalis* promotes the production of autoantibodies, driving the autoimmune response. However, some research report conflicting findings, indicating that *P. copri* levels might be influenced by external factors, such as diet or medication, which complicates their potential role of microbiota as a primary driver of RA. In addition to *P. copri*, other microbial imbalances, such as an overabundance of *Clostridium perfringens* and a reduction in *Faecalibacterium prausnitzii*, have been observed in RA patients, further implicating specific bacterial species in disease pathogenesis [69]. Microbial infection is also considered as a crucial inducer of RA. Most recently, a strain of *Subdoligranuluma*, isolated from both RA patients and patients at risk, has been demonstrated to be capable of colonizing in healthy mice and reproducing this human disease in a mouse model [73].

As we discussed in Section 2, the gut microbiome modulates intestinal epithelial barrier function and immune responses, playing important roles in the development and progression of chronic inflammatory diseases. Specifically, three microbiome-related mechanisms contribute to RA pathogenesis: (1) leaky gut epithelia mediated systemic inflammation, (2) molecular mimicry and autoantibody production, and (3) immune cell modulation via microbial metabolites. In RA patients, the integrity of the gut barrier has been shown to be compromised and accompanied by increased levels of LPS in the bloodstream, which is a potent stimulator of systemic inflammation [70]. However, questions about whether elevated LPS levels are a primary cause or a consequence of RA have been raised since the increased gut permeability might result from systemic inflammation rather than driving it [74]. Certain dietary or lifestyle factors also potentially impact both microbiome and gut barrier function, adding complexity to this relationship [75]. In addition, dysbiotic gut microbiota can stimulate the production of autoantibodies against self-antigens, further perpetuating autoimmune responses in RA [76]. Chriswell et al. [73], recently isolated a strain of *Subdoligranuluma* (named *S. didolesgii*) from RA patients. Mice orally administered this bacterium developed systemic autoantibodies and T cells that attacked the joints. Interestingly, mice with B or CD4+ T cell depletion not only halted intestinal immune responses but also eliminated detectable clinical disease [73]. The hypothesis is that the bacteria may express certain proteins that closely resemble human joint antigens, thereby confusing the immune system to attack both [77]. However, the role of molecular mimicry in RA remains a topic of debate, with some researchers arguing that the evidence is still inconclusive and that alternative mechanisms, such as direct microbial effects on immune cells, may also be important. This phenomenon is further supported by findings that RA patients exhibit increased levels of antibodies against bacterial antigens, indicating a direct link between gut bacteria and the immune response in RA. Furthermore, reduced levels of butyrate in RA patients have been linked to increased proinflammatory activity and reduced regulatory T-cell populations, highlighting the importance of microbial metabolites in disease modulation [78]. Nevertheless, the specific role of butyrate in RA remains under investigation, with some studies suggesting that its effects might vary depending on the context or patient population.

#### 3.1.3. Therapeutics of RA Targeting Gut Microbiome: Current and Future

Therapeutic modulation of the gut microbiota through probiotics, prebiotics, and dietary interventions has been proposed as a potential strategy for influencing RA-related outcomes [48]. For example, a high-fiber diet has been shown to increase the production of SCFAs, which enhance anti-inflammatory responses and potentially mitigate RA severity [49]. Clinical trials involving the administration of specific probiotic strains, such as *Lactobacillus casei*, have demonstrated improvements in disease activity scores and inflammatory markers in RA patients [79]. Another emerging approach is FMT, which has shown early signals of restoring gut microbial balance and alleviating RA symptoms in a preliminary study [50]. These approaches listed above aim to promote a healthy gut microbiome, thereby reducing systemic inflammation and potentially supporting joint health. Nevertheless, current evidence should be interpreted cautiously, as most studies are exploratory and lack large-scale validation.

The gut microbiome may also influence the response to RA therapies, as suggested by a recent retrospective, observational cohort study of 32 patients. Samples were collected at two timepoints (at least 6 months apart) during RA treatment. Upon analyzing the microbiomes, the study identified microbial patterns associated with the clinical treatment outcomes [80]. While these findings are intriguing, they remain preliminary and require validation in larger, prospective cohorts before they can inform clinical practice. Nonetheless, they highlight the potential importance of understanding the precise mechanisms by which the gut microbiota influences RA and provide other opportunities for RA management.

Future research should focus on several specific areas to advance our understanding of the role of the gut microbiota in disease mechanisms and therapeutic strategies. First, longitudinal studies are needed to track changes in the gut microbiota composition before and after the onset of RA to identify potential microbial biomarkers for early diagnosis. Second, detailed investigations into how specific gut bacteria or microbial metabolites influence systemic inflammation and joint pathology in RA are crucial. Targeted interventions, such as customized probiotic or prebiotic treatments based on individual microbiota profiles, should be explored to assess their efficacy in reducing disease activity and improving patient outcomes. Additionally, mechanistic studies elucidating the interactions between the gut microbiota and autoreactive immune responses in RA will be pivotal in developing novel microbiome-modulating therapies.

### 3.2. Inflammatory Bowel Disease (IBD)

#### 3.2.1. Gut Microbiome and IBD Pathogenesis

IBD represents a group of chronic inflammatory disorders of the GI tract, predominantly comprising Crohn’s disease (CD) and ulcerative colitis (UC) [81]. These conditions are characterized by dysregulated immune responses against the commensal gut microbiota in genetically susceptible individuals [52] (Table 2). There were approximately 4.9 million cases of IBD worldwide in 2019 [51]. The etiology of IBD is multifactorial, involving complex interactions among genetic predispositions, environmental factors, and dysbiosis of the gut microbiota [81]. Patients with IBD often exhibit increased bacterial translocation from the gut lumen into mesenteric lymph nodes and systemic circulation. As discussed in Section 2.1, this translocation is partly due to compromised intestinal barrier function and can lead to systemic immune activation [82]. Unlike other chronic inflammatory diseases, this phenomenon is particularly prominent in IBD, reflecting the severe disruption of gut barrier integrity as a hallmark of IBD.

Recent research has highlighted the importance of host-microbe interactions in the pathogenesis of IBD, suggesting that specific microbial patterns may serve as biomarkers for disease prediction and progression. However, some researchers caution that despite promising findings, the reliability of microbial signatures as consistent biomarkers remains debated due to variability across studies. In individuals with IBD, alterations in the composition and function of the gut microbiome are commonly observed, characterized by decreased microbial diversity, altered microbial composition and metabolites, and reduced abundance of beneficial bacteria such as *F. prausnitzii* and *Bifidobacterium* species [83]. These alterations are often accompanied by an increased presence of potentially pathogenic bacteria, such as *Escherichia coli* and *C. difficile*, which can exacerbate intestinal inflammation [84,85]. Some research implies that pathogenic bacteria may be opportunistic rather than primary drivers of inflammation, and their role in IBD pathogenesis may be secondary to underlying dysbiosis [86]. We have highlighted several IBD featured bacterial strains in Section 3.2.2.

Moreover, dysbiosis of the gut microbiota may influence the efficacy of therapeutic interventions for IBD and contribute to disease progression. Understanding the intricate interplay between the gut microbiota and IBD pathogenesis is essential for the development of novel therapeutic strategies that aim to restore microbial homeostasis and alleviate intestinal inflammation in affected patients.

#### 3.2.2. Certain Gut Microbes and IBD

In IBD, there is a notable increase in mucosa-associated bacteria, particularly in the ileum of CD patients and the colon of UC patients. These bacteria, such as *Ruminococcus gnavus*, can degrade the mucosal layer, exposing the epithelium to microbial antigens and triggering inflammatory response [87]. The presence of these bacteria near the epithelial surface is a distinctive feature contributing to the pathogenesis of IBD.

Adherent-invasive *E. coli* (AIEC) strains have been specifically associated with CD. These bacteria can adhere to and invade intestinal epithelial cells, survive and replicate within macrophages, and induce a strong proinflammatory response [88]. AIECs can form biofilms on the intestinal mucosa, which protect them from host immune responses and antibiotics, leading to the chronic inflammation characteristic of CD.

A decreased abundance of beneficial microbes such as *F. prausnitzii*, known for their anti-inflammatory properties, result in the loss of regulatory signals that normally help maintain immune homeostasis. Concurrently, the overgrowth of pathogenic bacteria such as *E. coli* can stimulate proinflammatory pathways [89]. These pathogens can penetrate the mucus layer, adhere to the intestinal epithelium, and induce the production of inflammatory cytokines such as TNF-α, thereby exacerbating the inflammatory cascade [90]. Nonetheless, the direct causative relationship between these pathogens and IBD-related inflammation is still under scrutiny, as some studies suggesting that inflammation may also arise from other aspects of microbial dysbiosis.

#### 3.2.3. Current Microbiota-Targeted Interventions for IBD

Emerging therapeutic approaches targeting the gut microbiota are gaining momentum in the management of IBD. Current microbiota-related interventions such as prebiotics, probiotics, symbiotics, herbal medicines, and FMT, aim to reduce intestinal dysbiosis, enhance mucosal barrier integrity, and mitigate inflammation. These strategies have shown encouraging signals, particularly in the management of conditions like active UC, pouchitis, and cases complicated by *C. difficile* infection. Here, we focus on discussing FMT, probiotics, prebiotics and dietary interventions.

FMT involves the transfer of fecal material from a healthy donor to the GI tract of a patient with IBD to restore gut microbial balance. Preliminary results from early clinical trials indicate that FMT can induce remission in some patients with UC [53,54], although its efficacy in patients with CD remains variable [55]. The long-term safety and optimal protocols for FMT in IBD patients remain unresolved, underscoring the need for larger, controlled studies.

Probiotics are live microorganisms intended to confer health benefits. They have been investigated for their ability to enhance gut barrier function and modulate immune responses. Prebiotics are nondigestible food components that promote the growth of beneficial bacteria and are being explored as adjunctive treatments for IBD. By combining prebiotic intake with dietary modifications, such as incorporating anti-inflammatory foods and reducing processed foods, individuals can further support gut health and mitigate systemic inflammation [91]. Specific probiotic strains, such as *Bifidobacterium* and *Lactobacillus* species, have shown promise in reducing inflammation and maintaining remission, particularly in UC patients [92]. However, the effects of probiotics can be strain-specific and may not be universally effective for all patients. Prebiotics, such as inulin and fructo-oligosaccharides (FOS), can increase the growth of beneficial bacteria and improve gut health, potentially offering another avenue for modulating the microbiota in IBD [93,94].

Dietary interventions play a crucial role in shaping the gut microbiota and influencing IBD progression. Diets, such as the low-FODMAP (standing for fermentable oligosaccharides, disaccharides, monosaccharides, and polyols) diet, exclusive enteral nutrition (EEN), and the Mediterranean diet, have been associated with reducing inflammation, promoting microbial diversity, and supporting the growth of beneficial bacteria [95]. Personalized dietary interventions based on individual microbiota profiles may represent a promising area of research, aiming to optimize the therapeutic benefits in IBD patients.

The efficacy of biologic therapies, such as anti-TNF agents, is also influenced by the gut microbiota composition. Some studies suggest that patients with a greater diversity of gut microbiota and a greater abundance of specific beneficial bacteria respond better to these therapies [96]. The mechanisms underlying these observations are not fully understood but may involve the role of the microbiota in modulating immune responses and drug metabolism. Personalized microbiota profiling may predict patient responses to biologic treatments and eventually inform therapeutic strategies, but this concept requires rigorous validation.

#### 3.2.4. Future Directions for Microbial Therapies of IBD

In the realm of IBD, future research should focus on identifying specific microbial signatures associated with disease onset and flare-ups through longitudinal studies. This approach could facilitate the development of diagnostic tools that leverage microbial profiles for early diagnosis and ongoing monitoring of IBD. Additionally, understanding the functional roles of key microbial species and their metabolites in modulating intestinal inflammation, barrier function, and immune responses is essential. Such insights can drive the development of targeted therapies, including personalized probiotics and dietary interventions. Clinical trials are needed to rigorously evaluate the efficacy of microbiota-targeted treatments, such as FMT and tailored prebiotic and probiotic regimens, in improving patient outcomes and sustaining prebiotic remission. Exploring the interactions between the gut microbiota and genetic or environmental factors related to IBD may also provide novel insights into disease mechanisms and guide the development of individualized treatment strategies.

### 3.3. Psoriasis

#### 3.3.1. General Understanding of Psoriasis Pathogenesis

Psoriasis is a chronic, autoimmune inflammatory skin disorder characterized by the rapid proliferation of keratinocytes, resulting in erythematous, scaly plaques primarily on the elbows, knees, scalp, and lower back. Globally, it affects more than 100 million people, with the prevalence ranging between 0.09 and 11.43% in countries [56]. A key feature of psoriasis is the activation of DCs and T helper 17 (Th17) T cells in the skin, accompanied by elevated levels of proinflammatory cytokines—particularly the IL-17/IL-23 axis [97]. This complex and multifactorial condition involves a combination of genetic predispositions, environmental triggers, and immune dysregulation (Table 2). Common triggers include stress, infections, comorbid illness such as diabetes, and physiological conditions like obesity.

#### 3.3.2. Gut Microbiome and Psoriasis Pathogenesis

Although the relationship between gut microbiome and psoriasis remains under investigation, emerging research suggested that gut dysbiosis may contribute to the disease’s onset and progression [98,99]. Clinical studies have consistently revealed that individuals with psoriasis exhibit altered gut microbiota compared to healthy controls, including reduced microbial diversity, shifts in specific taxa, and an imbalance between beneficial and pathogenic species [100]. Specifically, reductions in beneficial genera such as *Bifidobacterium*, *Akkermansia* and *Lactobacillus*, and increases in proinflammatory microbes such as *Ruminococcus torques* and *R. gnavus*, have been observed [99,101,102]. However, argument exists about whether these microbial alterations are a cause or consequence of psoriasis. Two recent studies using human fecal samples showed that the changes in microbiota observed in psoriasis patients do not meet the criteria for dysbiosis [103,104]. The authors proposed that these changes may reflect a microbial response to subclinical intestinal inflammation, with uncertain implication for the host.

Despite ongoing debates, the existence of a gut-skin axis is increasingly supported by growing evidence, highlighting its role in modulating inflammatory skin conditions of psoriasis. Disruptions in gut microbiota have been associated with increased intestinal permeability, allowing bacterial products to trigger systemic inflammation that worsens skin lesions in psoriasis. For instance, a case–control study of dermatology patients indicated that elevated levels of interleukin-1α in response to specific gut microbiota alterations were associated with psoriasis [105]. In preclinical therapeutic models, the combination of traditional psoriasis treatment such as methotrexate with probiotics *Bifidobacterium longum* has demonstrated synergistic benefits, including the preservation of intestinal barrier function, reduction in proinflammatory cytokines, and rebalancing of Th17/Treg cell populations [106]. Moreover, altered levels of gut-derived metabolites, particularly SCFAs, have been found in psoriasis patients [107]. Supplementation of SCFAs in drinking water has shown potential in alleviating the skin thickening, reducing IL-17 level, and restoring the diversity of fecal microbiota in a preclinical animal model [108]. Altogether, understanding the role of the gut microbiota in psoriasis may lead to new avenues for therapeutic interventions aimed at restoring microbial balance and reducing inflammation.

#### 3.3.3. Therapeutics of Psoriasis Targeting Gut Microbiome: Current and Future

Current research is exploring the potential of modulating the gut microenvironment through microecological interventions, such as probiotics and FMT, as adjunctive strategies for psoriasis management. Several small-scale clinic trials conducted globally have investigated the use of various probiotics as adjuvant therapies for psoriasis (Table 2). These studies generally reported favorable outcomes, including improvements in skin condition, reductions in inflammatory responses, and shift in microbiome composition [109,110,111,112]. However, not all findings were positive. One study found no significant differences between the control group and the group receiving *Lactobacillus rhamnosus* as adjuvant therapy [113], implying the therapeutic efficacy of probiotics may vary depending on the specific strains and disease manifestation. To date, no clinical trial involving FMT for psoriasis has been reported in the PubMed database. The only available report is a case study published in Chinese but with an English abstract, which described the improved skin condition of a patient with comorbid irritable bowel syndrome (IBS) and psoriasis following two FMT treatments [114]. In preclinical research, FMT from healthy human donors was shown to protect against Treg/Th17 imbalance and to modulate both gut and skin microbiota in a mouse model of psoriasis [115]. The therapeutic role of the gut microbiota in psoriasis has also been highlighted in studies examining traditional Chinese medicine. Shenling Baizhu powder (SLBZP) has been shown to alleviate psoriasis symptoms in mice by modulating gut microbiota and lipid metabolism [116].

Future research on the role of the gut microbiota in psoriasis should aim to identify specific microbial dysbiosis patterns that correlate with disease severity and response to treatment. Longitudinal studies investigating changes in microbial composition before, during, and after flare-ups could help in understanding how the gut microbiota influences psoriatic inflammation and skin barrier dysfunction. Additionally, research should focus on elucidating the mechanisms by which specific bacterial strains and their metabolites modulate systemic immune responses, particularly in relation to Th17 cell activation, which is central to psoriasis pathogenesis. This knowledge could drive the development of microbiota-based interventions, such as targeted probiotics or postbiotics, that specifically reduce inflammatory pathways involved in psoriasis. Moreover, clinical trials are needed to evaluate the efficacy of gut microbiota-targeted therapies, including dietary modifications and FMT, in achieving sustained clinical remission in psoriasis patients. Finally, exploring how gut-skin axis interactions are influenced by environmental factors and genetic predispositions may provide deeper insights into individualized therapeutic approaches for psoriasis.

### 3.4. Systemic Lupus Erythematosus (SLE)

#### 3.4.1. Disease Pathogenesis

SLE is a chronic autoimmune disease characterized by widespread inflammation and tissue damage affecting multiple organ systems, including the skin, joints, kidneys, heart, lung and nervous system. About 3.41 million people worldwide are estimated to have SLE [59]. The pathogenesis of SLE involves a complex interplay of genetic, environmental, and hormonal factors that lead to immune system dysregulation. Multiple genes are believed to be related to the susceptibility of individuals [117] (Table 2). Several environmental factors like UV radiation, infection, and stress, can trigger the disease in susceptible individuals [118,119]. Key features of SLE include the production of a wide array of autoantibodies, particularly antinuclear antibodies (ANAs) [120], and the formation of immune complexes that deposit in tissues [121]. Thus, in SLE patients, the immune system mistakenly attacks healthy tissues, resulting in chronic inflammation and damage [122]. Recent studies have identified specific autoantibodies, such as anti-Smith antibodies and anti-dsDNA antibodies, which are highly specific to SLE and correlate with disease activity. However, there is ongoing debate about whether these autoantibodies are direct drivers of disease pathology or merely markers of underlying immune dysregulation.

#### 3.4.2. Gut Microbiome and SLE

Studies have implied a potential link between the dysbiosis of gut microbiota and the pathogenesis of SLE [123]. Significant alterations in the gut microbiome of patients with SLE compared with healthy controls have been highlighted in recent research. These alterations are characterized by a reduction in microbial diversity and changes in the relative abundance of specific bacterial taxa and microbial metabolites [124]. Notably, studies have reported decreased levels of beneficial bacteria such as *Firmicutes* and *Bifidobacterium*, alongside an increase in potentially pathogenic bacteria such as *R. gnavus* and *Proteobacteria*. *R. gnavus*, which is associated with disease flares, can produce proinflammatory polysaccharides that exacerbate systemic inflammation [125]. Conversely, some evidence suggests that while *R. gnavus* is prevalent in SLE patients, its direct role in disease exacerbation may be less clear, with other microbial or host factors potentially playing a more significant role [126]. Additionally, the increase in *Proteobacteria* is concerning, as these bacteria include various pathogenic species that can elicit strong immune responses, further aggravating the autoimmune condition [127]. However, the specific pathogenic potential of *Proteobacteria* and their precise role in SLE remain subjects of ongoing investigation, with some researchers suggesting that these bacteria might act more as opportunistic pathogens than as primary drivers of disease. The microbial changes in SLE are not consistent across all studies, suggesting that the relationship between specific microbial taxa and SLE severity might be more complex or influenced by factors such as geographical location or diet [128]. Meanwhile, some researchers question the extent to which gut dysbiosis contributes to SLE, arguing that other factors, such as genetic predisposition or environmental triggers, may play more central roles.

Current research on the mechanisms by which gut dysbiosis contributes to the pathogenesis and progression of SLE remains limited. Similarly to RA, gut dysbiosis in SLE may trigger inflammation through three main pathways: (1) systemic inflammation mediated by leaky gut epithelium, (2) molecular mimicry and autoantibody production, and (3) immune cell modulation via microbes and their metabolites. Evidence of a leaky gut in SLE includes elevated serum LPS levels in patients [129,130] and translocation of specific bacteria, such as *Enterococcus gallinarum* [131] and *Lactobacillus reuteri* [132], to internal organs in both SLE animal models and patients. These microbes can enhance plasmacytoid dendritic cells (pDCs) activity and activate the interferon signaling pathway. Additionally, SLE patients exhibited increased levels of serum antibodies against gut bacteria and autoantibodies [133,134], indicating systemic immune activation likely driven by gut barrier dysfunction. Unlike RA, which primarily affects the joints, SLE involves immune cell activation that targets multiple organs throughout the body.

#### 3.4.3. Microbial Therapeutic Strategies/Interventions and Future Directions

The emerging evidence suggests a potential link between gut dysbiosis and SLE, promoting exploration of microbiota-targeted strategies primarily at the preclinical stage. Approaches including FMT, probiotics, prebiotics, and dietary modifications have been proposed to restore microbial balance and possibly influence disease outcomes [135]. The first clinical trial to evaluate the safety of FMT in SLE patients was recently done in China [60]. Patients responding to FMT showed reduced inflammatory responses and improved *Firmicutes*: *Bacteroidetes* ratios. This exploratory trial provides positive evidence to support FMT as a potential treatment for SLE in the future. However, given the small number of participants enrolled and the exploratory in nature, findings should be interpreted cautiously and require confirmation in larger, controlled studies.

Probiotics have been investigated for their potential to enhance the gut barrier integrity, reducing systemic inflammation, and modulating immune responses in various studies. A clinical trial reported in Indonesia has explored the use of a specific probiotic mixture (60% *Lactobacillus helveticus*, 20% *Bifidobacterium infantis*, and 20% *Bifidobacterium bifidum*) to reduce systemic IL-6 level and SLE disease activity and alter the composition and functions of gut microbiota [61]; yet, another trial conducted in Iran demonstrated that the daily regimen of 200 g of probiotic yogurt containing *L. rhamnosus* and *B. bifidum* for 13 weeks did not improve SLE disease activity index compared to untreated control [136]. These contrasting results highlight strain-specific effects and underscore the need for more rigorous trials. Adding prebiotics, such as inulin and FOS [62], into the diet may selectively promote the growth of beneficial gut bacteria and potentially alleviate SLE symptoms, but clinical evidence remains limited.

The primary goal in SLE management is preventing irreversible organ damage, which requires identifying key molecular contributors to disease progression. Early diagnosis and interventions alleviate disease symptoms, slow down disease progression and improve patient outcomes. Omics technologies, including genomics, metabolomics, and interactomics, have the potential to identify microbial biomarkers to assist in better understanding SLE pathogenesis and progression [137]. Precise studies are still needed to reveal effective microbiota-targeted therapies. Replenishment of beneficial microbes and metabolites into the gut environment of SLE patients may also be evaluated as adjuvants to improve the efficacy of conventional treatments and maintain the balance of disease activity and host immune responses.

### 3.5. Asthma

#### 3.5.1. Evidence of Gut Microbiome Association

Asthma is a chronic inflammatory disorder of the airways characterized by recurrent episodes of wheezing, dyspnea, chest tightness, and coughing. Affecting millions of people globally, asthma imposes substantial healthcare burdens and reduces quality of life. Asthma triggers are broadly categorized into allergic stimuli including pollen, dust mites, animal dander, and non-allergic stimuli such as air pollutants, respiratory infections, medications, and weather extremes.

While traditional research has focused on genetic predisposition, environmental triggers, and immune dysregulation, emerging evidence underscores the gut microbiota as a key modulator of immune responses and asthma development [138]. While adult-onset asthma is often associated with severe phenotypes and irreversible lung function decline, the role of microbiome alterations in adults remains poorly understood. In contrast, childhood asthma has been strongly linked to disruptions in early-life gut microbiota colonization. The first 100 days of life represent a period of heightened immune plasticity and microbial colonization. Perturbations during this window, such as perinatal antibiotic use, cesarean delivery, formula feeding, or prematurity, delay microbiota diversification and deplete anti-inflammatory microbial metabolites (e.g., SCFAs), consequently elevating asthma risk [139,140]

The low abundance of specific genera, for example, *Lachnospiraceae*, *Faecalibacterium* and *Lachnospira*, have been reported in the gut microbiota of children with high risk. Enrichment of beneficial strains, *Bifidobacterium* and *Lactobacillus*, via breast feeding play a role in preventing asthma development [141]. It is possible that these strains are capable of metabolizing human milk oligosaccharides (HMOs) into acetate and lactate which are critical for Treg cell development and suppression of inflammation [142]. However, colonization by *C. difficile* or fungal species (e.g., *Candida albicans*) in infancy is linked to later asthma onset [143].

#### 3.5.2. The Gut-Lung Axis in Asthma Pathogenesis

The gut-lung axis encompasses a bidirectional, immunologically mediated dialogue between the gastrointestinal and respiratory systems. Both the gut and lung microbiome are integral to maintaining immune homeostasis, and their dysregulation has been implicated in chronic respiratory diseases, such as athsma and chronic obstructive pulmonary disease (COPD) [144]. In asthma, this axis modulates pulmonary inflammation through multiple interconnected mechanisms, with early-life gut dysbiosis emerging as a critical driver of immune misprogramming and metabolic imbalance that elevates disease risk. The key mechanisms of the gut-lung axis in asthma include anti-inflammatory metabolites and immune regulation, gut dysbiosis and inflammation, and enhanced IgE production. Gut microbiota-derived SCFAs, such as acetate, propionate, and butyrate, are significantly reduced in the sera of asthma patients [145]. Metabolites originated from gut may enter systemic circulation and modulate the migration of immune cells within the airway. SCFAs can interact with the host G-protein coupled receptor GPR41 to inhibit NF-kB signaling and allergic airway inflammation [146,147]. SCFAs have also been shown to shape immune cell differentiation in the lungs, by promoting Treg differentiation and suppressing Th2-driven responses, thereby restoring the Th1/Th2 balance [148,149]. Again, those metabolites can enhance gut barrier integrity to prevent microbial translocation and systemic inflammation. In contrast, gut dysbiosis is correlated with distinct inflammatory profiles. The composition of the gut microbiota has also been associated with different types of inflammation, such as eosinophilic or neutrophilic inflammation, which could contribute to the development of distinct asthma phenotypes [150]. Antibiotic administration has been shown to aggravate asthma by disrupting the gut microbiota and intestinal barrier function in asthma mouse models [151]. Specific bacteria, including *Prevotella* and *Bacteroides*, have been implicated in activating mechanisms associated with inflammation, potentially linking disturbances in the gut microbiota to asthma pathogenesis. Particular gut microbes also regulate the migration of immune cells. *Proteobacteria* in the gut microbiota have been shown to modulate type 2 innate lymphoid cells (ILC2s) migration to the lung for host defense by regulating cytokines such as IL-33 [152]. Additionally, dysbiosis can drive type 2 inflammation by induction of cytokines, such as IL-4 and IL-13, that enhance IgE class switching and production, which contribute to the inflammatory cascade [153].

#### 3.5.3. Gut Microbiome-Based Therapeutics

Microbial therapies, primarily probiotics, have been investigated in both pediatric and adult asthma populations. But current evidence remains preliminary and heterogeneous. In clinical trials, probiotics have been applied either as a primary or adjuvant therapy to manage asthma symptoms, or as prophylactic intervention to prevent asthma development in high-risk infants. Over the past decade, the most frequently studied probiotic strains in clinical trials have belonged to the *Lactobacillaceae* family [140,154,155,156,157,158], and the *Bifidobacterium* genus [154,157,159,160]. However, the efficacy of these probiotics remains inconclusive due to inconsistent outcomes across those studies. These discrepancies may be attributed to variations in study design, including differences in specific probiotic strains, dosages, treatment durations, and follow-up parameters. Additionally, many studies involved small cohorts, which may have contributed to the variability in results.

Other specific strains such as *Ligilactobacillus salivarius*, *Clostridium butyricum*, *Limosilactobacillus reuteri* and *Saccharomyces boulardii* have been explored in individual trials [159,161,162,163]. Notably, *L. salivarius* plus *Bifidobacterium breve* were evaluated in a large cohort of over 400 pediatric patients in Italy, where the probiotic treatment reduced asthma exacerbations frequency by more than one third. Treatment with *S. boulardii* showed partial improvement in pulmonary function and modulated IgE and IL-5 levels in patients. In preclinical study, *S. boulardii* was found to alleviate symptoms by upregulate methyltransferase-like 3 in an N6-methyladenosine-dependent manner [162]. But these findings should be interpreted as early signals rather than established efficacy.

Although FMT has not yet been applied in clinical asthma management, preclinical models suggest it may effectively reduce respiratory inflammation and alter the gut microbial composition [164]. Furthermore, dietary interventions such as soluble fiber have demonstrated potential in modulating gut microbiota and reducing airway inflammation in asthma patients [154].

#### 3.5.4. Future Directions

Future research should prioritize the identification of microbial signatures associated with airway inflammation and bronchial hyperresponsiveness in asthma. Longitudinal studies tracking the early-life gut microbiome are essential to clarifying its role in shaping immune tolerance and disease susceptibility. Moreover, monitoring shifts in both gut and airway microbiomes may reveal biomarkers to predict disease progression and dissect in-depth microbiome correlated pathogenesis. Understanding the gut-lung axis, particularly how gut dysbiosis contributes to Th2-skewed immune responses, remains a critical area of exploration. Additionally, investigations of microbial metabolites and their capacity to modulate airway immune responses may reveal novel therapeutic pathways. Both precision probiotics and prebiotic formulations have the potential to reduce disease severity and prevent exacerbations and are worth further study in clinical trials. Assessing the efficacy of microbiome-targeted therapies in clinical trials with larger cohorts is necessary. Lastly, studies on the impact of environmental factors (such as antibiotic exposure and dietary patterns) on gut microbiota and asthma phenotypes will be instrumental in developing personalized prevention and treatment strategies.

### 3.6. Vasculitis

#### 3.6.1. Gut Microbiome and Vasculitis Pathogenesis

Vasculitis is a rare autoimmune disease, encompassing a group of disorders, such as Kawasaki disease (KD), giant cell arteritis, Behçet’s disease (BD), and Henoch–Schönlein purpura (HSP) that are characterized by inflammation of the blood vessels. The prevalence of vasculitis varies wildly by type, with an estimated incidence of fewer than 50 cases per million people for a single type [66] (Table 2). This inflammation can lead to vessel wall damage, impaired blood flow, and subsequent injury to organs and tissues. Vasculitis may involve arteries, veins, and capillaries in various organ systems, including the gastrointestinal tract, and presents with a wide array of symptoms. The specific etiology of vasculitis remains incompletely understood, but it is generally believed to arise from abnormal immune responses, where the body’s immune system mistakenly attacks its own blood vessels.

In recent years, growing evidence has implicated the gut microbiome in the development and progression of systemic vasculitis through its influence on host immune responses and vascular inflammation. Numerous studies have reported correlations between gut microbiota alterations and the onset of vasculitis [165,166]. For instance, transplantation of fecal microbiota from HSP patients into germ-free rats induced heightened visceral sensitivity and increased intestinal motility, indicating that gut dysbiosis may occur and correlate to the onset of the disease in HSP patients [167]. Other studies have also demonstrated that certain bacterial species within the gut microbiota, such as *Clostridium* and *Bacteroides*, regulate systemic immune responses crucial in vasculitis [168]. Moreover, specific bacterial changes, such as an increased abundance of *Proteobacteria* and reduced abundance of *Actinobacteria*, have been linked to organ involvement in vasculitis, particularly in IgA vasculitis [169]. In the context of KD, an immune-mediated vasculitis, prevalence of *Fusobacteria*, *Shigella* and *Streptococcus* might contribute to KD pathogenesis [170]. The intestinal microbiota composition of KD patients has also been implicated in the development of cardiovascular lesions associated with the disease. Research has identified specific intestinal commensals as regulators in a mice model, suggesting that a gut microbiota-cardiovascular inflammation axis is involved in KD pathogenesis [171]. Recent studies have also explored the association of the gut microbiota between children with IgA vasculitis and their mothers, suggesting a potential intergenerational influence of the maternal microbiota on the development or progression of IgA vasculitis in children [172]. These findings emphasize the role of the gut microbiota composition in immune-related vasculitis and highlight the potential transgenerational impact of the microbiota on disease susceptibility. Distinct patterns in microbial composition may help differentiate between types of vasculitis, despite the presence of shared flora, and may contribute to the pathogenesis of specific disease phenotypes [173].

The research about the correlation between gut microbiome and vasculitis was initiated in the past decade. Therefore, the precise mechanisms of how gut microbiome alteration trigger various phenotypes of vasculitis remain unclear. Generally, a gut dysbiosis-mediated reduction in beneficial microorganisms and anti-inflammatory metabolites can compromise the intestinal barrier, allowing bacterial components such as LPS to enter the bloodstream, triggering the activation of systemic inflammatory pathways and contributing to vasculitis pathogenesis [174]. The interaction between bacterial metabolites and TLRs on immune cells in blood vessels can lead to the activation of the NF-κB pathway, a key driver of inflammatory responses in vasculitis [175]. This sequence of events underscores the complex relationships among gut dysbiosis, increased intestinal permeability, bacterial translocation, and the development of vasculitis, highlighting the critical role of the gut microbiota in maintaining overall health.

#### 3.6.2. Microbial Interventions and Future Direction

While much of the existing literature has focused on correlations between gut dysbiosis and vasculitis, relatively few studies have evaluated microbial interventions for therapeutic purposes. Some progress has been made in preclinical models using dietary interventions [176], probiotics [177,178], and bacterial metabolites [179], with limited clinical validation. For example, administration of *Eubacterium rectale* in a BD mouse model reduced frequencies of CD83+ cells, since the frequencies of CD83+ cells were also significantly increased in patients with active BD. The *E. rectale* treatment significantly increased NK1.1+ cells populations with the improvement of symptoms [177]. Another study demonstrated that the supplementation with *C. butyricum* improved gut barrier integrity, increased abundance of the SCFAs-producing bacterial population and attenuated inflammation in a KD mouse model [178]. These results are promising but remain confined to animal studies.

To date, only one randomized clinical trial has investigated dietary modulation of gut microbiota to reduce inflammation in vasculitis patients. The study reached its endpoint in 2023, but results have not yet been published [180].

Metagenomic profiling studies have identified specific bacterial populations, such as Gram-positive bacteria, as key drivers of gut microbiome shifts associated with autoimmune conditions such as IgA vasculitis (IgAV), offering insights into potential targets for disease management [181]. Advances in sequencing and computational methods may enable the discovery of additional bacterial strains that serve as predictive biomarkers for disease onset and progression.

Further clinical trials are warranted to evaluate the potential efficacy of specific probiotic strains and prebiotic compounds in alleviating symptoms of vasculitis with the aim of reducing flare-ups and informing future therapeutic strategies. Certain strains of *Lactobacillus* and *Bifidobacterium* have demonstrated immunomodulatory effects in experimental settings, though evidence from human studies remains limited and variable. Tregs play a pivotal role in maintaining immune tolerance and preventing the immune system from attacking self-antigens [182], which is a fundamental problem in vasculitis. More robust clinical data are needed to substantiate the therapeutic potential of probiotics in vasculitis management.

Furthermore, emerging evidence supports the potential use of mesenchymal stem cells (MSCs) to modulate gut microbiota-immune system interactions [183]. Understanding the precise mechanisms by which the gut microbiota influences vascular inflammation and immune regulation will be key to developing targeted microbiota-based therapies for vasculitis. Longitudinal studies are needed to monitor shifts in the gut microbiota composition in vasculitis patients undergoing different treatments. By targeting the gut microbiota, it may be possible to modulate the immune response, reduce vascular inflammation, and mitigate the symptoms of vasculitis. This innovative approach highlights the importance of the gut–immune axis in the pathogenesis of vasculitis and offers a promising strategy for managing the disease beyond conventional immunosuppressive therapies.

## 4. Conclusions

Patients with chronic inflammatory diseases often exhibit dysregulated immune responses despite the inflammation manifesting in different parts of the body. Both genetic predisposition and environmental factors are believed to contribute to disease development. In recent decades, technological advances have uncovered the pivotal role of the gut microbiome in the onset and progression of various chronic inflammatory conditions. Although each disease presents its own unique characteristics, dysbiosis is a common feature among patients, even though the specific gut microbiota patterns may differ across diseases. While it remains unclear whether alterations in gut microbiome composition are a driving force or a consequence of the disease, it is evident that the gut microbiota interacts closely with immune cells residing in the intestinal epithelium. These interactions are essential for immune system development, immune modulation through microbial metabolites, and the maintenance of gut epithelial barrier integrity. Deciphering the intricate relationship between the gut microbiota and the immune system is fundamental to understanding the pathogenesis of chronic inflammatory diseases. This knowledge has also opened new therapeutic avenues for preventing and managing chronic inflammatory diseases by leveraging the ability of gut microbiota to influence systemic inflammation and restore immune balance.

While significant progress has been made, several challenges persist. Establishing causality, standardizing methodologies, and conducting longitudinal studies are crucial for better elucidating the temporal dynamics of changes in the gut microbiota. As the field has evolved, the integration of multiomic approaches and the exploration of microbial indicators such as short-chain fatty acids (e.g., butyrate) and specific bacterial species will provide deeper insights into the role of the gut microbes in disease modulation. Moreover, cross-disease analyses and/or studies on the comorbidity of two or more chronic inflammatory diseases are needed to identify both unique and shared microbial profiles among different chronic inflammatory conditions. While FMT may be broadly applied to restore a healthy gut microbiome, discovering specific common microbial strains or metabolites that are either positively or negatively associated with inflammation could lead to the development of microbial adjuvants or engineered strains to enhance conventional therapies. Although probiotic interventions have been most frequently investigated in chronic inflammatory diseases and may have shown some potential in alleviating disease symptoms as adjuvant therapies, varied regimen designs (e.g., dose, probiotic strains, duration, etc.) may lead to discrepancies in results even with same strains. Thus, large-scale clinical trials, with more precise regimen design are still needed to evaluate their long-term efficacy in specific diseases. At the same time, given the complexity of both disease mechanisms and the gut environment, personalized microbiota-based therapies may offer a promising strategy to restore microbial balance and reduce inflammation.

In summary, unraveling the complexities of gut-immune interactions hold great promise for advancing precision medicine. Continued research in this area aims to improve therapeutic outcomes and enhance the quality of life for individuals affected by these pervasive conditions.

## Figures and Tables

**Figure 1 microorganisms-13-02516-f001:**
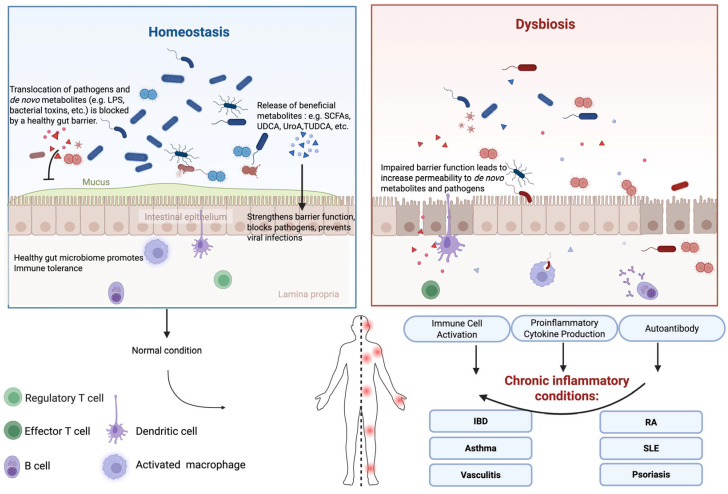
Overview of Gut Microbiome-Driven Immune and Inflammatory Modulation.

**Table 1 microorganisms-13-02516-t001:** Glossary.

Terms	Definition
MICROBIOTA	Microorganisms, composed of bacteria, fungi, virus, protozoa and archaea, inhabiting a defined environment.
MICROBIOME	Generally, microbiota, its genes, gene products and activities in niches in a habitat.
DYSBIOSIS	There is no consensus that defines dysbiosis despite a high frequency of usage in microbiome studies. It is often described as a state, in which alterations to the microbiota of hosts and its functional components may be correlated with undermined host immunity and increasing susceptibility to diseases. Dysbiosis usually features: (i) impaired microbial diversity; (ii) loss of beneficial commensal bacteria; (iii) thriving of pathogens.
DISEASE DRIVER	Causal factors that initiate or promote disease development or progression.
BIOMARKER	A subset of markers that are objectively measurable and evaluated as indicators of biological process, pathogenic processes, or responses to therapeutic interventions.
PATHWAY	Biological mechanisms or networks through which disease processes unfold, often involving multiple molecular interactions.

**Table 2 microorganisms-13-02516-t002:** Representative features and current microbial therapies for the selected chronic inflammatory diseases.

Disease	Estimated Global Prevalence	Primary Etiology	Autoimmune Features	Major Host Susceptibility Genes	Gut Microbiome-Targeted Therapies Tested Clinically	Supporting Literature
Rheumatoid Arthritis (RA)	~17·6 million ([47] Black, R. J. et al., 2023)	Joint-targeted autoantibodies	Yes	*HLA* alleles, *PTPN22*, *PADI4*, *STAT4*, *CTLA4*, *IL2RA*, etc.	High-fiber diet; Probiotics (e.g., *L. casei*); FMT	[45,46,47,48,49,50]
Inflammatory Bowel Disease (IBD)	~4.9 million ([51] Wang, R. et al., 2023)	Aberrant immune response to gut commensals	Yes	*NOD2*, *IL23R*, *HLA*, etc.	FMT; Probiotics (e.g., *Bifidobacterium*, *Lactobacillus*); Prebiotics (e.g., FOS); Diet (e.g., low-FODMAP, EEN, Mediterranean diet)	[51,52,53,54,55]
Psoriasis	at least 100 million ([56] WHO, 2016)	Skin-targeted inflammation	Yes	*NFKB1*, *ZFYVE28*, *IL23R*, *IL12B*, etc.	Probiotics (e.g., *B. infantis*, *Bacillus* genus, etc.); Prebiotics (e.g., fructooligosaccharides, SXRG84, etc.)	[56,57,58]
Systemic Lupus Erythematosus (SLE)	~3.41 million ([59] Tian, J. et al., 2023)	Multi-organ autoantibody production	Yes	*TYK2*, *STAT1*, *IRF5*, *STAT4*, etc.	FMT; Probiotics (e.g., *L. helveticus*, *B. infantis*, *B. bifidum*); Diet (e.g., inulin and FOS)	[59,60,61,62,63]
Asthma	~260 million ([64] Oh, J. et al., 2025)	Allergic/immune dysregulation in the respiratory tract	No	*ORMDL3*, *IL33*, *TSLP*, etc.	Probiotics (e.g., *L. salivarius*, *B. brev*, *C. butyricum,* etc.); Diet (e.g., inulin)	[64,65]
Vasculitis	varies widely by type; average < 50 cases per million people for a single type ([66] Watts, R.A. et al., 2022)	Vessel-targeted autoimmunity	Yes	KD-associated genes: *ITPKC*, *CASP3*, *ORAI1, BLK*, etc. BD-associated genes *IL-10A*, *CPVL*, *STAT4*, *TNFAIP3*, etc.	Dietary modulation	[66,67,68]

## Data Availability

No new data were created or analyzed in this study.

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
