# Peer review of "Gut Microbiome and Immune System Crosstalk in Chronic Inflammatory Diseases: A Narrative Review of Mechanisms and Therapeutic Opportunities"

_microorganisms, 2025, doi:10.3390/microorganisms13112516_

Round 1

Reviewer 1 Report (Previous Reviewer 1)

Comments and Suggestions for Authors

Overal, the authors carefully justified their stance regarding major comments and changing the title also adds to the clarification the type of review and which kind of research might expected as basis. The structure of the narrative review respects the main choices for disorders targeted by the authors and is thoroughly explored. However it still would be recommendable add some minor changes for a better understanding by the reader.

Table 1: I encourage reformulate the information added to the column Estimated Global Prevalence. Please, consider adding instead:

Last Name of first author, First letter of first name of first author et al., Year of the reference

Eg.: Feng, L. et al., 2019.

Figure 1: I encourage the authors to enlarge the figure and to improve the resolution to 1200dpi.

Author Response

Table 1: I encourage reformulate the information added to the column Estimated Global Prevalence. Please, consider adding instead:

Last Name of first author, First letter of first name of first author et al., Year of the reference

Eg.: Feng, L. et al., 2019.

 Response: Corrected.

Figure 1: I encourage the authors to enlarge the figure and to improve the resolution to 1200dpi.

Response: We appreciate the reviewer for the thoughtful suggestion. The figure at 600dpi provides high resolution suitable for zooming in and out. However, after we embedded it into the Word file, the resolution appears to have been reduced. To address this issue, we will upload the original high-resolution figure separately to the system.

Reviewer 2 Report (Previous Reviewer 2)

Comments and Suggestions for Authors
  1. "Our goal is to synthesize the latest research on the role of gut microbiome ..." - The word 'latest' should be replaced with 'recent'.
  2. Table 2 glossary can be better integrated into the narrative and should appear at the beginning rather than the end of the manuscript.
  3. Please correct numbering of references 141-183.
Comments on the Quality of English Language

Wordsmithing required.

Author Response

1. "Our goal is to synthesize the latest research on the role of gut microbiome ..." - The word 'latest' should be replaced with 'recent'.

Response: Corrected.

2. Table 2 glossary can be better integrated into the narrative and should appear at the beginning rather than the end of the manuscript.

Response: We appreciate the reviewer for the thoughtful suggestion. Some of these terms have already been described in the narrative (For exemple, see Page 2, Line 44-46; Page 3, Line 85-87). Additionally, we have revised the glossary table, now presented as Table 1, and cited it earlier in the text (Page 2, Line 47).

3. Please correct numbering of references 141-183.

Response: Corrected.

Reviewer 3 Report (Previous Reviewer 3)

Comments and Suggestions for Authors

Critical analysis of the authors’ responses
The authors reply courteously and make some useful clarifications (e.g., that the paper is explicitly a narrative review and not intended to be systematic). However, the responses only partially resolve  core concerns about (i) methodological transparency and (ii) novelty relative to the group’s earlier work. Key opportunities to strengthen rigor without converting the paper into a full systematic review remain underused.

What the authors addressed well:
a).Scope and intent clarified. They now state unambiguously that this is a narrative review and say they added clarifying language in the title/intro to avoid implying PRISMA-style methods. In the manuscript itself, the title and framing indeed present it as a narrative review and the search is described as PubMed-based with recent-years English literature and specified keyword families, which matches their response. 
b). Organization and readability. The piece remains well structured (mechanisms - disease sections - therapeutics), and disease-specific subsections are coherent and didactic (e.g., RA, IBD, psoriasis, SLE, asthma, vasculitis).
c). Tempering of claims. In several disease sections, the text acknowledges conflicting or inconclusive clinical evidence (e.g., probiotics/FMT in psoriasis; limited, exploratory FMT in SLE), which helps reduce overstatement.

What remains insufficient or unconvincing:
a). Methodological transparency (still thin for a “comprehensive” synthesis). Even for a narrative review, journals typically expect minimum transparency: databases (beyond PubMed), time window, inclusion heuristics, how conflicting evidence was prioritized, and whether key meta-analyses/systematic reviews were preferentially weighted. The current Methods description (keywords, PubMed, 15-year English-only window) remains brief; there is no evidence table per condition, no explicit approach to weighing study quality, and no rationale for excluding/including conflicting trials. Adding a short “Evidence selection and weighting in this narrative review” box plus one compact evidence map/table per disease would markedly improve rigor without converting to PRISMA. The manuscript’s own description confirms a keyword search in PubMed, English language, and a 15-year window, but does not spell out selection or appraisal steps. 
b). Novelty vs. prior work is asserted, not demonstrated. The response argues that their 2022 paper was on CDI/live biotherapeutics (infectious) while this review covers non-infectious chronic inflammatory diseases; that distinction is valid. Still, substantial conceptual material (definitions of dysbiosis, SCFAs/bile acids, barrier and immune pathways, and the standard probiotic/FMT toolkit) appears again here. To dispel redundancy concerns, they should add an explicit “What is new compared with our 2022 review” paragraph and a brief comparative table (columns: concept/theme; coverage in 2022 CDI review; new expansion/insights here across RA/IBD/psoriasis/SLE/asthma/vasculitis). Without this, the claim of distinct contribution remains under-substantiated.
c). Critical appraisal depth varies across diseases. Some sections thoughtfully note negative/inconclusive trials (e.g., psoriasis, SLE), but others still lean descriptive. A concise “Clinical evidence snapshot” per disease (highest-level evidence first; include null/negative trials; indicate sample sizes and key limitations) would improve balance and reduce optimism bias.
d). Causality cautions should be more consistent. The manuscript intermittently flags the cause-vs-consequence dilemma of dysbiosis but could make this a unifying throughline with a dedicated box (“Causality caveats and ways forward: longitudinal cohorts, Mendelian randomization, gnotobiotic transfer, mechanistic readouts”).
e).Terminology and english style.The responses note edits, but the manuscript would still benefit from tightening (occasional repetition, long sentences, and a few typographical slips). A light editorial pass should fix these. Perform a language polish to tighten phrasing and remove remaining typos.

Actionable, non-PRISMA improvements the authors can still make now:
1). Add a brief “Evidence selection and weighting” paragraph (databases searched; date limits; preference for systematic reviews/meta-analyses where available; how contradictory findings were handled).
2). Insert one compact evidence table per disease (archetypal mechanisms + best human evidence + most robust interventional data + key null/negative trials + main gaps).
3). Provide a comparative novelty table vs. their 2022 CDI review to pre-empt self-overlap concerns.
4). Add a aausality and methods caveats box (cause vs consequence; phenotype heterogeneity; small trials; strain specificity; publication bias).

 Evaluation of the revised manuscript 
1).Is the work a significant contribution to the field? Useful cross-disease synthesis with accessible mechanistic framing. Contribution is moderate because conceptual territory overlaps standard microbiome-immunity narratives and prior group themes; adding explicit novelty and evidence maps would raise impact.
2). Is the work well organized and comprehensively described? Well structured and readable; however, “how the evidence was gathered and weighed” remains under-described for a broad, “comprehensive” narrative review. The manuscript itself presents as a narrative review and outlines a PubMed, English-only, recent-years keyword strategy, but details of selection/appraisal are minimal. 
3). Is the work scientifically sound and not misleading? Generally accurate mechanistic descriptions, with some good acknowledgment of conflicting/limited clinical data in certain sections. Still, critical appraisal depth and consistent treatment of negative evidence could be stronger; causality caveats should be foregrounded.
4). Are there appropriate and adequate references to related and previous work? References are ample and current across diseases; nonetheless, prioritizing high-level evidence (recent systematic reviews/meta-analyses) more explicitly would strengthen the synthesis.

Specific comments for authors:
a). Please add a short evidence selection/weighting paragraph (databases; time window; preference hierarchy; handling contradictions) to increase transparency for a narrative review. 
b). Provide concise evidence tables per disease (include null/negative trials and sample sizes; highlight where only preclinical data exist).
c). Add a novelty statement + comparative table vs. your 2022 review to dispel self-overlap concerns and highlight what is genuinely new here (multi-disease synthesis; updates since 2022; therapeutic pipelines beyond CDI).
d). Unify causality caveats in a dedicated box and propose concrete designs to advance causality (longitudinal birth cohorts for asthma; MR analyses; standardized FMT/probiotic strain trials with immune biomarkers).
e). Tighten therapeutic claims to clearly distinguish signal from established efficacy, and to flag heterogeneity/strain specificity and small study sizes.

Comments on the Quality of English Language

The English could be improved to more clearly express the research.
Recommend a light editorial pass to tighten long sentences, remove repetition, and fix minor errors.

Author Response

Critical analysis of the authors’ responses
The authors reply courteously and make some useful clarifications (e.g., that the paper is explicitly a narrative review and not intended to be systematic). However, the responses only partially resolve  core concerns about (i) methodological transparency and (ii) novelty relative to the group’s earlier work. Key opportunities to strengthen rigor without converting the paper into a full systematic review remain underused.

What the authors addressed well:
a).Scope and intent clarified. They now state unambiguously that this is a narrative review and say they added clarifying language in the title/intro to avoid implying PRISMA-style methods. In the manuscript itself, the title and framing indeed present it as a narrative review and the search is described as PubMed-based with recent-years English literature and specified keyword families, which matches their response. 
b). Organization and readability. The piece remains well structured (mechanisms - disease sections - therapeutics), and disease-specific subsections are coherent and didactic (e.g., RA, IBD, psoriasis, SLE, asthma, vasculitis).
c). Tempering of claims. In several disease sections, the text acknowledges conflicting or inconclusive clinical evidence (e.g., probiotics/FMT in psoriasis; limited, exploratory FMT in SLE), which helps reduce overstatement.

Response: We thank the reviewer for the summary.

What remains insufficient or unconvincing:
a). Methodological transparency (still thin for a “comprehensive” synthesis). Even for a narrative review, journals typically expect minimum transparency: databases (beyond PubMed), time window, inclusion heuristics, how conflicting evidence was prioritized, and whether key meta-analyses/systematic reviews were preferentially weighted. The current Methods description (keywords, PubMed, 15-year English-only window) remains brief; there is no evidence table per condition, no explicit approach to weighing study quality, and no rationale for excluding/including conflicting trials. Adding a short “Evidence selection and weighting in this narrative review” box plus one compact evidence map/table per disease would markedly improve rigor without converting to PRISMA. The manuscript’s own description confirms a keyword search in PubMed, English language, and a 15-year window, but does not spell out selection or appraisal steps. 

Response: As this manuscript is explicitly framed as a narrative review rather than a systematic review, we believe that adding an “Evidence Selection and Weighting” box and detailed evidence tables would shift the scope toward a systematic approach, which is not the intent of this work. Our goal is to provide a broad, integrative synthesis rather than a structured evidence appraisal. We agree that transparency is important. The current methodology description has clearly stated the databases searched, time window, language restrictions, and keyword families, which we believe meet the expectations for narrative reviews.

b). Novelty vs. prior work is asserted, not demonstrated. The response argues that their 2022 paper was on CDI/live biotherapeutics (infectious) while this review covers non-infectious chronic inflammatory diseases; that distinction is valid. Still, substantial conceptual material (definitions of dysbiosis, SCFAs/bile acids, barrier and immune pathways, and the standard probiotic/FMT toolkit) appears again here. To dispel redundancy concerns, they should add an explicit “What is new compared with our 2022 review” paragraph and a brief comparative table (columns: concept/theme; coverage in 2022 CDI review; new expansion/insights here across RA/IBD/psoriasis/SLE/asthma/vasculitis). Without this, the claim of distinct contribution remains under-substantiated.

Response: We appreciate the reviewer’s perspective regarding the two reviews. However, we respectfully disagree that including such a table would enhance our manuscript. The gut microbiome is a foundational concept relevant across numerous disease contexts (for exp. metabolic disorders, oncology, neuroscience, infectious diseases, etc.), and research in this area is rapidly evolving. Comprehensive summaries of recent studies and developments within each disease field are essential to provide readers with an up-to-date understanding. These reviews, which address the microbiome in different disease settings, do not represent conceptual overlap. Additionally, comparing our previous study with the current work falls outside the scope of this manuscript and would not contribute to its improvement.

c). Critical appraisal depth varies across diseases. Some sections thoughtfully note negative/inconclusive trials (e.g., psoriasis, SLE), but others still lean descriptive. A concise “Clinical evidence snapshot” per disease (highest-level evidence first; include null/negative trials; indicate sample sizes and key limitations) would improve balance and reduce optimism bias.

Response: We appreciate the reviewer’s suggestion. All available clinical studies including those with inconclusive findings are discussed in the text and listed in Table 2 (previously Table 1). While we agree that summarizing high-level evidence can be valuable, the very limited number of human trials for certain conditions (e.g., vasculitis, as noted on Page 17, Lines 746–757) makes it impractical to include discussions about negative/inconclusive studies for these sections. This reflects the current evidence landscape rather than any bias in our review.

d). Causality cautions should be more consistent. The manuscript intermittently flags the cause-vs-consequence dilemma of dysbiosis but could make this a unifying throughline with a dedicated box (“Causality caveats and ways forward: longitudinal cohorts, Mendelian randomization, gnotobiotic transfer, mechanistic readouts”).

Response: We are sorry that we may misunderstand the reviewer’s point. Following the editor’s guidance, we have drawn an abstract graphic to illustrating the crosstalk between the gut microbiome and the conditions discussed in the manuscript. Additionally, Figure 1 has been modified to better align with the overall theme of the review. We hope those modifications can address the reviewer’s concern and enhance clarity for readers.

e). Terminology and english style.The responses note edits, but the manuscript would still benefit from tightening (occasional repetition, long sentences, and a few typographical slips). A light editorial pass should fix these. Perform a language polish to tighten phrasing and remove remaining typos.

Response: We thank the reviewer for the suggestion. We have polished the language. Please see all the revisions in red in the context.

Actionable, non-PRISMA improvements the authors can still make now:
1). Add a brief “Evidence selection and weighting” paragraph (databases searched; date limits; preference for systematic reviews/meta-analyses where available; how contradictory findings were handled).

Response: Again, this is a narrative review. The current methodology description has clearly stated the databases searched, time window, language restrictions, and keyword families, which we believe meet the expectations for narrative reviews.

2). Insert one compact evidence table per disease (archetypal mechanisms + best human evidence + most robust interventional data + key null/negative trials + main gaps).

Response: Thank you for the suggestion. However, our manuscript is structured as a narrative review, and these elements (best human evidence, interventional data, null trials, and gaps) are already integrated within each disease section. Creating additional tables or boxes would be redundant and duplicate content and significantly alter the narrative format. Additionally, whether dysbiosis is a cause or consequence of disease is still debated. We have provided a dedicated section discussing the overarching mechanisms, which we believe is more informative and aligned with the narrative review format. We believe the current structure provides clarity while maintaining the flow of discussion. Therefore, we respectfully decline this suggestion.

3). Provide a comparative novelty table vs. their 2022 CDI review to pre-empt self-overlap concerns.

Response: We appreciate the reviewer’s perspective regarding the two reviews. However, we respectfully disagree that including such a table would enhance our manuscript. The gut microbiome is a foundational concept relevant across numerous disease contexts (for exp. metabolic disorders, oncology, neuroscience, infectious diseases, etc.), and research in this area is rapidly evolving. Comprehensive summaries of recent studies and developments within each disease field are essential to provide readers with an up-to-date understanding. These reviews, which address the microbiome in different disease settings, do not represent conceptual overlap. Additionally, comparing our previous study with the current work falls outside the scope of this manuscript and would not contribute to its improvement.

4). Add a aausality and methods caveats box (cause vs consequence; phenotype heterogeneity; small trials; strain specificity; publication bias).

Response: Thank you for your comment. However, we believe that the issues you mentioned, such as causality versus consequence, phenotype heterogeneity, small trial limitations, strain specificity, and publication bias, are already addressed within the narrative of our review in the context of each disease. Adding a separate “Causality and Methods Caveats” box would be redundant and may disrupt the flow of a narrative review. Our goal is to provide an integrated discussion rather than isolated summaries. Therefore, we respectfully disagree that including such a box would improve the manuscript.

 Evaluation of the revised manuscript 
1).Is the work a significant contribution to the field? Useful cross-disease synthesis with accessible mechanistic framing. Contribution is moderate because conceptual territory overlaps standard microbiome-immunity narratives and prior group themes; adding explicit novelty and evidence maps would raise impact.
2). Is the work well organized and comprehensively described? Well structured and readable; however, “how the evidence was gathered and weighed” remains under-described for a broad, “comprehensive” narrative review. The manuscript itself presents as a narrative review and outlines a PubMed, English-only, recent-years keyword strategy, but details of selection/appraisal are minimal. 
3). Is the work scientifically sound and not misleading? Generally accurate mechanistic descriptions, with some good acknowledgment of conflicting/limited clinical data in certain sections. Still, critical appraisal depth and consistent treatment of negative evidence could be stronger; causality caveats should be foregrounded.
4). Are there appropriate avnd adequate references to related and previous work? References are ample and current across diseases; nonetheless, prioritizing high-level evidence (recent systematic reviews/meta-analyses) more explicitly would strengthen the synthesis.

Response: We thank the reviewer for the evaluations.

Specific comments for authors:
a). Please add a short evidence selection/weighting paragraph (databases; time window; preference hierarchy; handling contradictions) to increase transparency for a narrative review. 

Response: Please see Page 2&3, Line 65-82.

b). Provide concise evidence tables per disease (include null/negative trials and sample sizes; highlight where only preclinical data exist).

Response: Thank you for the suggestion. We appreciate the intent to provide a concise summary. However, our manuscript is structured as a narrative review, and these elements are already integrated within each disease section. Creating additional tables or boxes would be redundant and duplicate content and significantly alter the narrative format. Again, our goal is to provide an integrated discussion rather than isolated summaries. We believe the current structure provides clarity while maintaining the flow of discussion. Therefore, we respectfully decline this suggestion.

c). Add a novelty statement + comparative table vs. your 2022 review to dispel self-overlap concerns and highlight what is genuinely new here (multi-disease synthesis; updates since 2022; therapeutic pipelines beyond CDI).

Response: We appreciate the reviewer’s perspective regarding the two reviews. However, we respectfully disagree that including such a table would enhance our manuscript. The gut microbiome is a foundational concept relevant across numerous disease contexts (for exp. metabolic disorders, oncology, neuroscience, infectious diseases, etc.), and research in this area is rapidly evolving. Comprehensive summaries of recent studies and developments within each disease field are essential to provide readers with an up-to-date understanding. These reviews, which address the microbiome in different disease settings, do not represent conceptual overlap. Additionally, comparing our previous study with the current work falls outside the scope of this manuscript and would not contribute to its improvement.

d). Unify causality caveats in a dedicated box and propose concrete designs to advance causality (longitudinal birth cohorts for asthma; MR analyses; standardized FMT/probiotic strain trials with immune biomarkers).

Response: Thank you for the suggestion. We agree that causality is an important consideration and have discussed these caveats throughout the manuscript. However, our review is intended as a narrative synthesis of current research rather than a methodological roadmap. Adding a dedicated box and proposing specific study designs would shift the manuscript toward a prescriptive format, which is beyond its intended scope. We believe our integrated discussion of causality limitations and future directions provides sufficient context for readers.

e). Tighten therapeutic claims to clearly distinguish signal from established efficacy, and to flag heterogeneity/strain specificity and small study sizes.

Response: Thank you for the suggestion. We agree that therapeutic claims should be presented cautiously. In our revision, we have ensured that statements about interventions clearly distinguish preliminary signals from established efficacy and explicitly note limitations such as heterogeneity, strain specificity, and small sample sizes. Please see the revisions in red in the context.

This manuscript is a resubmission of an earlier submission. The following is a list of the peer review reports and author responses from that submission.

Round 1

Reviewer 1 Report

Comments and Suggestions for Authors

 Lines 23-30: "This review focuses on six representative chronic inflammatory diseases, including rheumatoid arthritis, inflammatory bowel disease, psoriasis, systemic lupus erythematosus, asthma, and vasculitis, all of which are characterized by dysregulated immune responses and persistent inflammation. Our goal is to synthesize the latest research on the role of gut microbiome in the pathogenesis of these diseases listed above and provide insights into the development of microbiota-based therapies, particularly fecal microbiota transplant and probiotic interventions, for their treatment." There is information missing here, since that in the introduction the authors particularly refer to "We also discuss the therapeutic potential of microbiota-targeted intervention, including dietary modifications, probiotics, prebiotics, and fecal microbiota transplantation (FMT)."
Regarding the summary of the work that is well formulated above I kindly invite the authors to review their title for somethign along: "Gut Microbiome and Immune System Crosstalk in Six/Some Chronic Inflammatory Diseases: Mechanisms and Therapeutic Insights"
Lines 63-69: " This article aims to explore the intricate functions of the gut microbiome within the human body, with a particular emphasis on its pivotal role in the onset and progression of several major chronic inflammatory conditions. Specifically, we discuss RA, inflammatory bowel disease (IBD), psoriasis, systemic lupus erythematosus (SLE), asthma, and vasculitis, diseases that collectively affect millions of individuals worldwide and pose a substantial global health burden (Table 1)." Table 1 should be placed right after this sentence.
Lines 71-75: "To ensure relevance and currency, we reviewed articles written in English from the past 10-15 years, identified through the PubMed database (why only PubMed? and why aren't limitations explored and possible overcomes for the near future? using only 1 database for searching is quite limiting) using the keywords correlated to the topics of “microbiome”, “chronic inflammatory diseases”, “host immune response and gut microbes”, “host immune response and microbial metabolites” and the specific disease names listed above."  
Please, ensure the exact number of years you included in your literature search and justify the reason for so. Moreover, please include the exact search string that would allow reproducing the study. Though, it is not a systematic review, it would be welcome to include a flowchart of studies searched, found and rejected and selected (PRISMA flowchart). Also, it would help to refer inclusion and exclusion criteria for the studies considered in the study.
Did the search particularly include search terms related to the therapeutic potential of microbiota-targeted intervention that the authors refer as a point/goal of discussion for the manuscript?
Lines 91-93: "We defined additional terms including “disease driver”, “biomarker” and “pathway” in Table 2" . Table 2 should placed after this sentence, please.
Figure 1 would benefit of resolution improvement.
From Line 177 on: (130 references to correct) - Please, for every reference such as Hubert Hug et al. correct to Hubert Hug et al., [number of reference].

Author Response

Comment 1: Lines 23-30: "This review focuses on six representative chronic inflammatory diseases, including rheumatoid arthritis, inflammatory bowel disease, psoriasis, systemic lupus erythematosus, asthma, and vasculitis, all of which are characterized by dysregulated immune responses and persistent inflammation. Our goal is to synthesize the latest research on the role of gut microbiome in the pathogenesis of these diseases listed above and provide insights into the development of microbiota-based therapies, particularly fecal microbiota transplant and probiotic interventions, for their treatment." There is information missing here, since that in the introduction the authors particularly refer to "We also discuss the therapeutic potential of microbiota-targeted intervention, including dietary modifications, probiotics, prebiotics, and fecal microbiota transplantation (FMT)."
Regarding the summary of the work that is well formulated above I kindly invite the authors to review their title for somethign along: "Gut Microbiome and Immune System Crosstalk in Six/Some Chronic Inflammatory Diseases: Mechanisms and Therapeutic Insights"

Response 1:

(1) We thank the reviewer for pointing out the inconsistency between the abstract and the introduction regarding the therapeutic strategies discussed. We have revised the abstract to include “dietary modifications” and “prebiotics” alongside FMT and probiotics, reflecting the full scope of microbiota-targeted interventions covered in the manuscript (Page 1, Line 30-31).

(2) We appreciate the reviewer’s thoughtful suggestion regarding the manuscript title. However, we would like to maintain the broader framing of our original title to reflect the narrative nature of the review and its emphasis on synthesizing current knowledge across multiple disease areas. But, meanwhile, we are open to refining the title for clarity and impact. Based on the reviewer’s input, we propose the following revised title: “Gut Microbiome and Immune Crosstalk in Chronic Inflammatory Diseases: A Narrative Review of Mechanisms and Therapeutic Opportunities”. We hope this strikes a balance between specificity and the narrative scope of the review.

Comment 2: Lines 63-69: " This article aims to explore the intricate functions of the gut microbiome within the human body, with a particular emphasis on its pivotal role in the onset and progression of several major chronic inflammatory conditions. Specifically, we discuss RA, inflammatory bowel disease (IBD), psoriasis, systemic lupus erythematosus (SLE), asthma, and vasculitis, diseases that collectively affect millions of individuals worldwide and pose a substantial global health burden (Table 1)." Table 1 should be placed right after this sentence.

Response 2:

We thank the reviewer for the thoughtful suggestion. But if we have interpreted the comment correctly, the final formatting and structure of the manuscript including layout will be determined by the editorial office upon acceptance. We are happy to make any necessary adjustments at that stage in accordance with the journal’s guidelines.

Comment 3: Lines 71-75: "To ensure relevance and currency, we reviewed articles written in English from the past 10-15 years, identified through the PubMed database (why only PubMed? and why aren't limitations explored and possible overcomes for the near future? using only 1 database for searching is quite limiting) using the keywords correlated to the topics of “microbiome”, “chronic inflammatory diseases”, “host immune response and gut microbes”, “host immune response and microbial metabolites” and the specific disease names listed above."  

Response 3:

We appreciate the reviewer’s insightful observation regarding our literature search strategy. In this narrative review, we chose to focus on the PubMed database due to its comprehensive coverage of biomedical literature and its relevance to the topics discussed, including microbiome research and chronic inflammatory diseases. Our aim was to synthesize current knowledge rather than conduct a systematic review, which typically involves multiple databases and formal inclusion/exclusion criteria. We believe that PubMed sufficiently supported our objectives. We acknowledge the value of including multiple databases in systematic reviews and appreciate the reviewer’s suggestion for future work to consider broader search strategies.

Comment 4: Please, ensure the exact number of years you included in your literature search and justify the reason for so. Moreover, please include the exact search string that would allow reproducing the study. Though, it is not a systematic review, it would be welcome to include a flowchart of studies searched, found and rejected and selected (PRISMA flowchart). Also, it would help to refer inclusion and exclusion criteria for the studies considered in the study.

Response 4: 

(1) We appreciate the reviewer’s attention to the timeframe referenced in our literature search. We primarily searched literature in the past 15 years. But as noted in the manuscript, research on the role of microbiome in vasculitis has emerged in the past decade. To ensure consistency and avoid confusion, we have revised ‘10-15 years’ to ‘15 years’ (Page 2, Line 74).

(2) Thank you for your thoughtful suggestion regarding the inclusion of a PRISMA flow diagram. We would like to clarify that this manuscript is a narrative review as indicated in the title, which differs from a systematic review in its methodology and scope. Narrative reviews typically do not follow the structured protocol required for systematic reviews, such as predefined inclusion/exclusion criteria or formal study selection processes. While we conducted a comprehensive literature search to ensure the relevance and currency of the included studies, our approach was not designed to meet the formal requirements of a systematic review. Therefore, we believe that a PRISMA flow diagram may not be appropriate for this type of review. Instead, we have clearly described our search strategy and selection process in Page 2, Line 73-77 to maintain transparency.

Comment 5: Did the search particularly include search terms related to the therapeutic potential of microbiota-targeted intervention that the authors refer as a point/goal of discussion for the manuscript?

Response 5:

During the literature search, we did not include the therapeutic terms, such as FMT, probiotics, etc. in the initial keyword strategy. Instead, we identified the therapeutic approaches discussed in the manuscript based on their prominence and frequency within the literature retrieved using broader search terms. To clarify, we have added the statement in Page 2, Line 72-73.

Comment 6: Lines 91-93: "We defined additional terms including “disease driver”, “biomarker” and “pathway” in Table 2" . Table 2 should placed after this sentence, please.

Response 6:

We thank the reviewer for the thoughtful suggestion. But if we have interpreted the comment correctly, the final formatting and structure of the manuscript including layout will be determined by the editorial office upon acceptance. We are happy to make any necessary adjustments at that stage in accordance with the journal’s guidelines.

Comment 7: Figure 1 would benefit of resolution improvement.

Response 7: We have replaced it with one of higher resolution (from 300DPI to 600DPI).

Comment 8: From Line 177 on: (130 references to correct) - Please, for every reference such as Hubert Hug et al. correct to Hubert Hug et al., [number of reference].

Response 8: Corrected. See Page 5, Line 192 and Page 7, Line 268

Reviewer 2 Report

Comments and Suggestions for Authors
  1. Study title should indicate that this is a narrative review.
  2. In the introduction, the authors should explicitly state the purpose of the review, the review scope, and the intended contribution(s).
  3. Table 1 mixes global prevalence estimates drawn from heterogeneous sources and timeframes, without specifying reference years or population bases. Units should also be consistent (currently some in absolute millions, others per-million persons).
  4. For each section/condition, please categorize the findings more clearly, e.g., as preclinical, observational human, or clinical trial evidence. This is important as for asthma and vasculitis, evidence cited is predominantly preclinical, yet presented as if clinically established.
Comments on the Quality of English Language

Moderate edits needed. Writing tends to be verbose and convoluted.

Author Response

  1. Study title should indicate that this is a narrative review.

Response: We appreciate the reviewer’s thoughtful suggestion regarding the manuscript title. We propose the following revised title: “Gut Microbiome and Immune Crosstalk in Chronic Inflammatory Diseases: A Narrative Review of Mechanisms and Therapeutic Opportunities”. We hope this strikes a balance between specificity and the narrative scope of the review.

  1. In the introduction, the authors should explicitly state the purpose of the review, the review scope, and the intended contribution(s).

Response: We appreciate the reviewer’s thoughtful suggestion. Please see our description regarding the reviewer’s comment in:

  • Page 2, Line 63-73: “In light of these findings, there is growing interest in targeting the gut microbiota and its metabolites as a therapeutic strategy for chronic inflammatory diseases. This article aims to explore the intricate functions of the gut microbiome within the human body, with a particular emphasis on its pivotal role in the onset and progression of several major chronic inflammatory conditions. Specifically, we discuss RA, inflammatory bowel disease (IBD), psoriasis, systemic lupus erythematosus (SLE), asthma, and vasculitis, diseases that collectively affect millions of individuals worldwide and pose a substantial global health burden (Table 1). We also discuss the therapeutic potential of microbiota-targeted intervention, including dietary modifications, probiotics, prebiotics, and fecal microbiota transplantation (FMT) which were among the most frequently and prominently retrieved interventions when searching with the keywords listed below.”
  • Page 2, Line 78-80, “Our goal is to synthesize current knowledge of gut microbiome-related pathogenesis, highlight mechanistic insights, and identify therapeutic opportunities across those major disease areas.”

  1. Table 1 mixes global prevalence estimates drawn from heterogeneous sources and timeframes, without specifying reference years or population bases. Units should also be consistent (currently some in absolute millions, others per-million persons).

Response: We thank the reviewer for this important observation. We would like to clarify that in Table 1, the prevalence estimates for all conditions except vasculitis are presented in terms of absolute numbers (millions) of affected individuals globally. For vasculitis, we used per-million persons due to the variability in prevalence across its subtypes and the lack of consistent global estimates. To improve clarity, we have now added reference years within the table.

  1. For each section/condition, please categorize the findings more clearly, e.g., as preclinical, observational human, or clinical trial evidence. This is important as for asthma and vasculitis, evidence cited is predominantly preclinical, yet presented as if clinically established.

Response: We thank the reviewer for this thoughtful suggestion.

First, we would like to highlight that Table 1 summarizes therapeutic approaches that have been tested in human studies, as indicated in the Column titled “Gut microbiome-targeted therapies tested clinically”. To maintain clarity and avoid confusion, therapeutic strategies that have only been evaluated in preclinical models were intentionally excluded from the table. For example, FMT was not listed for asthma and vasculitis since it hasn’t been evaluated in any clinical trial or study.

Second, in the main text, we have clearly noted when evidence is primarily preclinical, particularly in the sections on asthma and vasculitis, where clinical data are limited. Below, we list the specific statements in the manuscript that are explicitly based on preclinical studies:

  1. Page 10, Line 453-457: “In preclinical therapeutic models, the combination of traditional psoriasis treatment such as methotrexate with probiotics Bifidobacterium longum has demonstrated synergistic benefits, including the preservation of intestinal barrier function, reduction of proinflammatory cytokines, and rebalancing of Th17/Treg cell populations [79].”
  2. Page 11, Line 458-460: “Supplementation of SCFAs in drinking water has shown potential in alleviating the skin thickening, reducing IL-17 level, and restoring the diversity of fecal microbiota in a preclinical animal model [81].”
  3. Page 11, Line 477-479: “In preclinical research, FMT from healthy human donors was shown to protect against Treg/Th17 imbalance and to modulate both gut and skin microbiota in a mouse model of psoriasis [158].”
  4. Page15, Line 657-659: “In preclinical study, boulardii was found to alleviate symptoms by upregulate methyltransferase-like 3 in an N6-methyladenosine-dependent manner [169].”
  5. Page 15, Line 660-662: “Although FMT has not yet been applied in clinic asthma management, preclinical models suggest it may effectively reduce respiratory inflammation and alter the gut microbial composition [171].”
  6. Page 16, Line 734-741: “Some progress has been made in preclinical models using dietary interventions [133], probiotics [134, 135], and bacterial metabolites [136]. Notably, administration of Eubacterium rectale in a BD mouse model reduced frequencies of CD83+ cells, since the frequencies of CD83+ cells were also significantly increased in patients with active BD. The rectale treatment significantly increased NK1.1+ cells populations with the improvement of symptoms [134]. Another study demonstrated that the supplementation with C. butyricum improved gut barrier integrity, increased abundance of the SCFAs-producing bacterial population and attenuated inflammation in a KD mouse model [135].”

Third, our intention was to present emerging therapeutic insights while maintaining transparency about the nature of the supporting evidence. We respectfully note that we are unsure why the reviewer felt that preclinical findings were presented as if clinically established, as we have made efforts to clearly distinguish the evidence type throughout the manuscript.

Reviewer 3 Report

Comments and Suggestions for Authors

The manuscript provides a broad and well-written synthesis of the interplay between the gut microbiome, immune responses, and chronic inflammatory diseases. It covers six representative conditions (RA, IBD, psoriasis, SLE, asthma, and vasculitis) and outlines both mechanistic aspects and therapeutic opportunities, including probiotics, prebiotics, dietary interventions, and FMT. The writing is clear and the review succeeds in emphasizing the biological plausibility of microbiome–immune interactions.

However, several critical methodological limitations reduce its originality and scientific robustness. Despite being framed as a comprehensive review, the authors state they identified articles in PubMed using keywords but do not present a structured search, inclusion/exclusion criteria, or evidence tables. This is inconsistent with best practices for systematic reviews (e.g., PRISMA). Instead, the paper functions as a narrative review with selective referencing, which risks bias and incompleteness. Additionally, the overlap with the group’s earlier work (Zhang, Y. et al., Gut Microbes 2022, DOI: 10.1080/19490976.2022.2052698) is evident: both manuscripts focus on microbiota reconstitution and therapeutic implications in chronic inflammatory contexts. While the new review broadens the scope to multiple diseases, many mechanistic explanations, concepts of dysbiosis, and therapeutic prospects are repeated with limited novel insight.

While the manuscript is informative and well-organized, it falls short of the standards expected for a systematic and original review. Strengthening methodological transparency, explicitly addressing novelty compared to prior publications by the same group, and integrating meta-analytical rigor would substantially improve the scientific contribution.

Specific Evaluation

1. Is the work a significant contribution to the field? The review consolidates important literature but lacks originality, given the overlap with the group’s prior article on live biotherapeutics against Clostridioides difficile. Contribution is moderate: broad in scope but not advancing new concepts or methods.

2. Is the work well organized and comprehensively described Structure is logical (general mechanisms → disease-specific sections → therapeutic strategies → conclusions). Tables and figures (e.g., overview of mechanisms, therapies per disease) are clear and informative. Still, the comprehensiveness is limited by the absence of a systematic methodology (search strategy, evidence grading).

3. Is the work scientifically sound and not misleading? The mechanistic explanations are scientifically accurate and supported by references. However, the absence of systematic review methodology and the occasional speculative associations (e.g., causality of dysbiosis vs. consequence of disease) reduce scientific rigor. Therapeutic claims (FMT, probiotics) are discussed but not critically appraised against negative or inconclusive trials.

4. Are there appropriate and adequate references to related and previous work References are abundant and up to date (many 2018–2024). However, key systematic reviews and meta-analyses are not systematically prioritized. Redundancy with the group’s 2022 paper suggests potential self-overlap and insufficient distinction between previous and current contributions.

Comparison with the group’s previouswork - Zhang, Y., Saint Fleur, A., & Feng, H. (2022). The development of live biotherapeutics against Clostridioides difficile infection towards reconstituting gut microbiota. Gut Microbes, 14(1), 2052698. DOI: 10.1080/19490976.2022.2052698. Both manuscripts define dysbiosis, microbiota composition, metabolites (SCFAs, bile acids, LPS), and therapeutic strategies (FMT, probiotics).The 2022 paper is narrowly focused on C. difficile infection and live biotherapeutics; the new manuscript expands to multiple chronic inflammatory diseases. Substantial conceptual repetition without clearly articulating what is new. The current manuscript should explicitly acknowledge and distinguish itself from their earlier review to avoid perceived redundancy.

Methodological assessment
The authors describe that they searched PubMed for articles in English within the past 10–15 years, using keywords. However, they do not report: number of articles retrieved and screened; inclusion/exclusion criteria; flowchart (as per PRISMA); any quality assessment of included studies. Without this, the review is narrative, not systematic, despite the scope suggesting a need for PRISMA-style methodology. This undermines reproducibility and transparency.

The authors should reframe the article explicitly as a sistematic review and  redesign the methodology following PRISMA (systematic search, eligibility, data extraction, and critical appraisal).

Best regards!

Author Response

The manuscript provides a broad and well-written synthesis of the interplay between the gut microbiome, immune responses, and chronic inflammatory diseases. It covers six representative conditions (RA, IBD, psoriasis, SLE, asthma, and vasculitis) and outlines both mechanistic aspects and therapeutic opportunities, including probiotics, prebiotics, dietary interventions, and FMT. The writing is clear and the review succeeds in emphasizing the biological plausibility of microbiome–immune interactions.

However, several critical methodological limitations reduce its originality and scientific robustness. Despite being framed as a comprehensive review, the authors state they identified articles in PubMed using keywords but do not present a structured search, inclusion/exclusion criteria, or evidence tables. This is inconsistent with best practices for systematic reviews (e.g., PRISMA). Instead, the paper functions as a narrative review with selective referencing, which risks bias and incompleteness. Additionally, the overlap with the group’s earlier work (Zhang, Y. et al., Gut Microbes 2022, DOI: 10.1080/19490976.2022.2052698) is evident: both manuscripts focus on microbiota reconstitution and therapeutic implications in chronic inflammatory contexts. While the new review broadens the scope to multiple diseases, many mechanistic explanations, concepts of dysbiosis, and therapeutic prospects are repeated with limited novel insight.

While the manuscript is informative and well-organized, it falls short of the standards expected for a systematic and original review. Strengthening methodological transparency, explicitly addressing novelty compared to prior publications by the same group, and integrating meta-analytical rigor would substantially improve the scientific contribution.

Response: We appreciate the reviewer’s detailed feedback and the opportunity to clarify the scope and methodology of our manuscript. We have carefully reply to the reviewer’s comments below. We hope these clarifications address the reviewer’s concerns and demonstrate the originality and relevance of our current manuscript.

Specific Evaluation

  1. Is the work a significant contribution to the field? The review consolidates important literature but lacks originality, given the overlap with the group’s prior article on live biotherapeutics against Clostridioides difficile. Contribution is moderate: broad in scope but not advancing new concepts or methods.

Response: We appreciate the reviewer’s feedback and the opportunity to clarify the scope and contribution of our current review.

While our group has previously published on live biotherapeutics targeting Clostridioides difficile infection (CDI), that work focused on an infectious disease with a distinct pathophysiology and clinical context. In contrast, the present review addresses chronic inflammatory diseases, which are non-infectious, immune-mediated conditions such as rheumatoid arthritis, IBD, psoriasis, SLE, asthma, and vasculitis. Although both reviews involve the gut microbiome, the disease mechanisms, therapeutic targets, and translational challenges are fundamentally different.

Our intention with this review is to synthesize current knowledge on gut microbiome-immune system crosstalk across major chronic inflammatory diseases and to highlight therapeutic opportunities that are emerging in this space. We believe this work contributes to the field by:

  • Providing a comparative overview across multiple disease areas.
  • Highlighting mechanistic insights into microbiome-immune interactions.
  • Summarizing human clinical evidence and identifying gaps for future research.

We respectfully disagree with the notion that prior work on CDI limits our ability to explore microbiome-targeted therapies in other disease contexts. On the contrary, we believe that expertise in one area can inform and enrich understanding in others, especially as microbiome science continues to evolve across disciplines.

  1. Is the work well organized and comprehensively described Structure is logical (general mechanisms → disease-specific sections → therapeutic strategies → conclusions). Tables and figures (e.g., overview of mechanisms, therapies per disease) are clear and informative. Still, the comprehensiveness is limited by the absence of a systematic methodology (search strategy, evidence grading).

Response: We would like to emphasize that this work is intended as a narrative review, not a systematic review. Our goal was to synthesize emerging insights across multiple chronic inflammatory diseases with a focus on gut microbiome-immune system interactions and therapeutic developments. As such, we did not follow a structured protocol involving predefined inclusion/exclusion criteria or a formal evidence table, which are standard for systematic reviews. We have now clarified this distinction in the title to avoid confusion.

  1. Is the work scientifically sound and not misleading? The mechanistic explanations are scientifically accurate and supported by references. However, the absence of systematic review methodology and the occasional speculative associations (e.g., causality of dysbiosis vs. consequence of disease) reduce scientific rigor. Therapeutic claims (FMT, probiotics) are discussed but not critically appraised against negative or inconclusive trials.

Response:

(1) This work is intended as a narrative review, not a systematic review or meta-analysis.

(2) We appreciate the reviewer’s suggestion and would like to clarify that our manuscript already includes multiple examples throughout that critically appraise the current evidence on microbiota-based therapies. For example:

Page 11, Line 466-473, we have acknowledged that while several small-scale clinical trials have reported favorable outcomes for probiotic interventions in psoriasis, at least one study found no significant therapeutic effect, highlighting variability in efficacy depending on strain and disease manifestation.

Page 11, Line 473-477, we note the absence of clinical trials for FMT in psoriasis, with only limited evidence from a single case report and preclinical studies, emphasizing the preliminary nature of current findings and the need for large-scale, rigorously designed trials to validate the therapeutic potential of probiotics and FMT in chronic inflammatory diseases.

Page 13, Line 556-560, we have acknowledged the exploratory nature and limited sample size of the first FMT trial in SLE patients.

Page 13, Line 562-567, we have described that same probiotic strains in different trials have inconclusive results.

There are more examples across the entire manuscript. These original discussions have already provided a nuanced and critical appraisal of the current evidence, aligning well with the reviewer’s suggestion.

  1. Are there appropriate and adequate references to related and previous work References are abundant and up to date (many 2018–2024). However, key systematic reviews and meta-analyses are not systematically prioritized. Redundancy with the group’s 2022 paper suggests potential self-overlap and insufficient distinction between previous and current contributions.

Response:

(1) Again, this work is intended as a narrative review, not a systematic review or meta-analysis.

(2) CDI is an infectious disease with a distinct pathophysiology and clinical context from chronic inflammatory diseases which are non-infectious, immune-mediated conditions.

Our 2022 article focused on treatments for CDI based on gut microbiome reconstitution. As CDI is an intestinal infectious disease, research about the role of gut microbiome in CDI is relatively extensive and well-established, with several FDA-approved microbiome-based therapies now available (REBYOTA, 2022 and VOWST, 2023).

In contrast, as discussed in the current manuscript, the microbiome field of chronic inflammatory diseases is still in an earlier stage of development regarding microbiome-targeted therapies. Although numerous studies have demonstrated associations between gut microbiota and the pathogenesis of these conditions (as discussed in our manuscript), the mechanistic understanding remains limited, and clinical applications are still emerging.

Our current review aims to summarize recent progresshighlight mechanistic insights, and inspire future therapeutic directions. Therefore, we respectfully disagree with the reviewer’s assessment that the article is significantly repetitive. We believe our two reviews serve distinct purposes and reflect the differing maturity levels of microbiome research in infectious versus chronic inflammatory diseases.

Comparison with the group’s previouswork - Zhang, Y., Saint Fleur, A., & Feng, H. (2022). The development of live biotherapeutics against Clostridioides difficile infection towards reconstituting gut microbiota. Gut Microbes, 14(1), 2052698. DOI: 10.1080/19490976.2022.2052698. Both manuscripts define dysbiosis, microbiota composition, metabolites (SCFAs, bile acids, LPS), and therapeutic strategies (FMT, probiotics).The 2022 paper is narrowly focused on C. difficile infection and live biotherapeutics; the new manuscript expands to multiple chronic inflammatory diseases. Substantial conceptual repetition without clearly articulating what is new. The current manuscript should explicitly acknowledge and distinguish itself from their earlier review to avoid perceived redundancy.

Response: We appreciate the reviewer’s observation. Again, we cannot agree with the reviewer on the claim of ‘conceptual repetition’ of the two reviews. While both manuscripts discuss foundational concepts such as dysbiosis, microbiota composition, and therapeutic strategies, the scope and focus of the two works are substantially different. Please see more of our responses to reviewer’s comment 1 and 4.

Methodological assessment

The authors describe that they searched PubMed for articles in English within the past 10–15 years, using keywords. However, they do not report: number of articles retrieved and screened; inclusion/exclusion criteria; flowchart (as per PRISMA); any quality assessment of included studies. Without this, the review is narrative, not systematic, despite the scope suggesting a need for PRISMA-style methodology. This undermines reproducibility and transparency.
The authors should reframe the article explicitly as a sistematic review and  redesign the methodology following PRISMA (systematic search, eligibility, data extraction, and critical appraisal).

Response: Thank you for your thoughtful suggestion regarding the inclusion of a PRISMA flow diagram. We would like to clarify one more time that this manuscript is a narrative review as indicated in the title, which differs from a systematic review in its methodology and scope. Narrative reviews typically do not follow the structured protocol required for systematic reviews, such as predefined inclusion/exclusion criteria or formal study selection processes. While we conducted a comprehensive literature search to ensure the relevance and currency of the included studies, our approach was not designed to meet the formal requirements of a systematic review. Therefore, we believe that a PRISMA flow diagram may not be appropriate for this type of review. Instead, we have clearly described our search strategy and selection process in Page 2, Line 73-77 to maintain transparency.
